# A novel Chua's based 2–D chaotic system and its performance analysis in cryptography

**Suresh Rasappan**[1☉], **Sathish Kumar Kumaravel**[2☉], **Aceng Sambas**[3,4‡], **Issam A. R. Moghrabi**[5,6‡]*, **Ibrahim Mohammed Sulaiman**[7,8‡], **Basim A. Hassan**[9‡]

1 Mathematics Section, University of Technology and Applied Sciences, Ibri, Sultanate of Oman, 2 Department of Mathematics, Vel Tech Rangarajan Dr. Sagunthala R & D Institute of Science and Technology, Avadi, Chennai, Tamil Nadu, India, 3 Faculty of Informatics and Computing, Universiti Sultan Zainal Abidin, Besut, Malaysia, 4 Department of Mechanical Engineering, Universitas Muhammadiyah Tasikmalaya, Tasikmalaya, Indonesia, 5 Computer Science Department, School of Arts and Sciences, University of Central Asia, Naryn, Kyrgyz Republic, 6 Department of Information Systems and Technology, Kuwait Technical College, Abu-Halifa, Kuwait, 7 Institute of Strategic Industrial Decision Modelling (ISIDM), School of Quantitative Sciences, Universiti Utara Malaysia, Sintok, Kedah, Malaysia, 8 Faculty of Education and Arts, Sohar university, Sohar, Oman, 9 College of Computer Science and Mathematics, University of Mosul, Mosul, Iraq

☉ These authors contributed equally to this work.
‡ AS, IARM, IMS and BAH also contributed equally to this work.
* i.moghrabi@ktech.edu.kw

**Data Availability Statement:** All relevant data are within the article.

**Funding:** The author(s) received no specific funding for this work.

## Abstract

In this study, the chaotic behavior of a second-order circuit comprising a nonlinear resistor and Chua's diode is investigated. This circuit, which includes a nonlinear capacitor and resistor among its components, is considered one of the simplest nonautonomous circuits. The research explores various oscillator characteristics, emphasizing their chaotic properties through bifurcations, Lyapunov exponents, periodicity, local Lyapunov region, and resonance. The system exhibits both stable equilibrium points and a chaotic attractor. Additionally, the second objective of this study is to develop a novel cryptographic technique by incorporating the designed circuit into the S-box method. The evaluation results suggest that this approach is suitable for secure cryptographic applications, providing insights into constructing a cryptosystem for images and text based on its complex behavior. Real-life data were analyzed using various statistical and performance criteria after applying the proposed methodology. These findings enhance the reliability of the cryptosystems. Moreover, The proposed methods are assessed using a range of statistical and performance metrics after testing the text and images. The cryptographic results are compared with existing techniques, reinforcing both the developed cryptosystem and the performance analysis of the chaotic circuit.

## Introduction

Researchers have infiltrated chaotic dynamic systems that delve into the intricacies of these systems, characterized by complications, unpredictable behavior, and sensitivity to initial

**Competing interests:** The authors have declared that no competing interests exist.

conditions. Notably, they have focused on identifying similar features in dynamic systems [1]. The chaotic nature of these systems is one of their most remarkable characteristics. Chaos is a phenomenon in dynamical systems theory in which a system displays chaotic behavior for a brief period before eventually stabilizing into a periodic or quasi-periodic state. This concept has been thoroughly investigated across various domains, including physics, mathematics, engineering, and biology.

In 1983, Celso Grebogi, Edward Ott, and James Yorke published a pioneering paper on chaos, which demonstrated that chaotic attractors in the Lorenz system could have a finite lifetime [2, 3]. Despite the unpredictability inherent in the Lorenz system, they showed that chaotic behavior can be predicted by studying the dynamics of unstable periodic orbits. This contribution introduces the concept of unstable periodic orbits and elucidates their significance in the emergence of chaos in dynamic systems.

Following the study by Grebogi, Ott, and Yorke, researchers have continued to explore the idea of transitory chaos in various systems, including mechanical systems, electronic circuits, and biological systems. A notable advancement in this field involves leveraging transient chaos to enhance the sensitivity of systems to minor perturbations, which is a property applied in diverse applications, including cryptography and secure communications.

Another significant stride in the field of transient chaos was the identification of the Ruelle-Takens scenario. This scenario illustrates a sequence of bifurcations that transpire as a dynamic system shifting from a stable fixed point to a chaotic attractor. The Ruelle-Takens scenario has been observed in several systems, including the logistic map, Henon map, and Lorenz system.

Recently, researchers have begun to explore the relationship between chaos and synchronization in complex networks. This research revealed that chaos can play a beneficial role in promoting synchronization in networks of coupled oscillators. Chaos has potential as a synchronization mechanism in fields such as power systems and communication networks.

In the realm of dynamical systems theory, extensive research has been conducted on the intriguing phenomenon of chaos. The concept of transitory chaos has led to significant advancements in our understanding of chaotic systems, including the Ruelle–Takens scenario [4] and the notion of unstable periodic orbits. Furthermore, the features of transient chaos have been applied in various real-world contexts, such as network synchronization and cryptography [5, 6].

This study focuses on a straightforward circuit with brief chaotic oscillations using a Chua's diode. In recent years, the design, circuit implementation, and analysis of oscillator qualities have emerged as important research areas. Oscillators that are prone to chaos have garnered interest owing to their potential applications. An oscillator generates alternating and repeating waves without any input. In this process, power signals are transformed into current signals, a fundamental characteristic frequently employed in electrical gadgets [7–10].

A chaotic oscillator is a mathematical representation of a system that is highly sensitive to the initial conditions and, evolves in an unpredictable and seemingly random manner. This contrasts with an optimization algorithm [11], a computational strategy designed to iteratively refine solutions to specific problems [12] with the aim of finding the optimal set of parameters or configurations [13]. Numerous natural and artificial systems, such as weather patterns, electrical circuits, and biological processes, exemplify this chaotic behavior [14, 15]. Chaotic oscillators have applications in various disciplines, including cryptography, in which they can generate reliable encryption keys. Moreover, they are employed in physics to analyze the behavior of complex systems and in biology to model the dynamic behaviors of populations of organisms [16].

Through various control strategies, such as feedback, time–delay feedback, and backstepping mechanisms [17], stabilization has emerged as a crucial tool for managing chaotic

behavior, including system perturbations and noise [18]. Chaos can be regulated through either the most likely technique [19, 20] or the sampled method [21]. This regulation is evident in the shifts in the positions of the points of the system in the phase view and the alterations in the attractor of the system [22–24].

In recent years, the prevalence of online shopping and consumption has significantly increased. Despite secure transmission, cybercrime is on the rise [25]. Modern data encryption methods rely on well-respected random number generation techniques used by cryptographers and other professionals [26, 27]. Chaotic models are commonly employed to generate random sequences because of their complex dynamics. Examples include [28–30]. The formation and subsequent utilization of a sequence leverages the effective features of a growing chaotic flow. Finally, a novel image cryptosystem utilizing the proposed random number technique is introduced. The efficiency of the proposed mechanisms was evaluated using various methods, with notable findings emphasizing their crucial characteristics for reliable cryptographic applications.

Chaotic oscillators have garnered significant research attention in recent years owing to their unique ability to generate complex, irregular signals with high sensitivity to initial conditions. Owing to these characteristics, chaotic oscillators are useful in various fields, including encryption. This study examines the effectiveness of transient chaotic oscillators in cryptography, focusing on their applications in encryption, decryption, and key generation.

To establish secure communication channels, encryption techniques have employed chaotic oscillators. The chaotic behavior of these oscillators can be exploited to generate encryption keys for encoding plaintext messages [31]. One such method is synchronization-based encryption, in which the synchronized states of two chaotic oscillators serve as encryption keys [32]. Another approach is chaos-based image encryption, which encrypts plaintext images using XORing pseudo-random sequences generated by chaotic oscillators with a plaintext image [33]. Some encryption methods have demonstrated promising resistance to various attacks, such as brute force attacks and statistical attacks [34].

Chaotic oscillators can also be used in the decryption process to extract the original plaintext message from encrypted ciphertext. In the chaos-based secure communication strategy, the chaotic oscillator used for encryption is employed for decryption. The receiver can obtain plaintext by synchronizing with the chaotic oscillator at the sender's end, as described in the 1996 generalized work by Kocarev [35]. Another strategy is the chaos-based cryptography technique in which a decryption key is generated by a chaotic oscillator and then used to reverse the encryption process. These decryption techniques have demonstrated their effectiveness in recovering the original plaintext from the ciphertext, while maintaining the confidentiality of the encrypted message.

Chaotic oscillators can also generate cryptographic keys that are essential for secure communication. The chaotic behavior of these oscillators can produce random, unpredictable sequences used for authentication, encryption, and decryption keys. The chaos-based key exchange strategy involves connecting two chaotic oscillators, with their synchronized states serving as cryptographic keys [36]. Another strategy is a chaos-based key agreement system, in which two communicating parties create a shared secret key using a chaotic oscillator [37]. These key generation algorithms have proven successful in producing secure and unpredictable cryptographic keys.

Chua's diode, a pivotal theoretical construct, plays a crucial role in chaotic oscillator circuits by exhibiting negative resistance and nonlinearity, thus enabling the creation of unpredictable oscillations [38]. Operational amplifiers and memristors are commonly employed in their implementation [39]. These diodes find applications in chaos generators used in cryptography,

communications, nonlinear circuit simulations, and modelling of various biological systems [40, 41].

This paper introduces a novel chaotic circuit design that integrates a Vonderpol oscillator and Chua's diode. The proposed system, inspired by the existing chaotic circuit literature, generates unpredictable oscillations with a straightforward implementation. The goals of this study were as follows:

- To develop an easy-to-use nonautonomous circuit that exhibits chaos and its qualitative characteristics, such as bifurcation and Lyapunov exponents.

- Furthermore, the designed circuits are used to encrypt and decrypt images and text.

- Algorithms for both text and image based cryptosystems have been developed using the *S-box* method and are based on tuned choatic circuits.

- The effectiveness of the proposed algorithms was evaluated using various tests, including statistical tests, histogram analyses, and key sensitivity analyses.

To address these goals, a mathematical model of the proposed oscillator is presented and detailed in Section 2. An exploration of whether bifurcation and Lyapunov exponent-invariant properties for the oscillator are discussed. Section 3 discusses the backstepping tactics. Sections 4 and 5 explore the cryptosystem scenario and offer conclusions for future developments.

## Circuit realization and chaos

A capacitor (C), inductor (L), nonlinear resistance (R), external periodic forcing, and a nonlinear element, notably the Chua's diode, make up the self-maintaining electrical circuit. As shown in Fig 1, they were all interconnected in series. This is a variation of the *Vander Pol* oscillator.

When Kirchoff's law is applied to this circuit, the voltage *v* across capacitor *C* and the current *i* via inductor $i_L$ are governed by two first-order non-autonomous differential Eq (1).

$$C\frac{dv_C}{dt} = i_L + fsin(\Omega t)$$

$$L\frac{di_L}{dt} = -v_C - Ri_L(v_C^2 - g(v_C)) \tag{1}$$

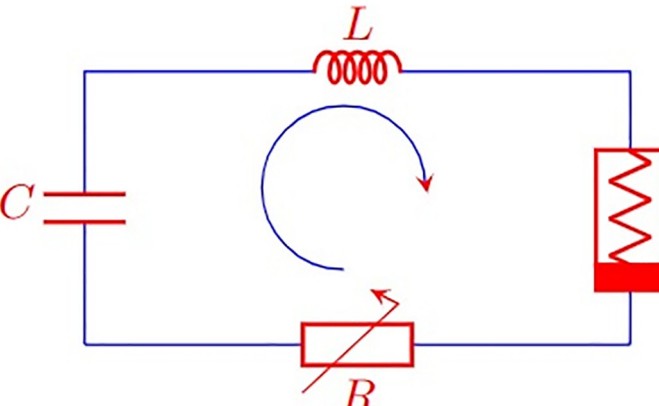

**Fig 1. Chaotic oscillator circuit diagram.**

where $f$ denotes the magnitude of an external periodic force, and is $\Omega$ its angular frequency. The Chua's diode's characteristic curve equation is given by (2):

$$g(v_C) = G_b v_C + 0.5(G_a - G_b)[|v_C + B_p| - |v_C - B_p|] \qquad (2)$$

By rescaling (1) as $v_c = xB_p$, $i_L = GyB_p$, $G = \frac{1}{R}$, $\omega = \frac{\Omega C}{G}$ and then, and by redefining $\tau$ as $t$, the following set of normalized Eq (3) is obtained:

$$\begin{aligned} \dot{x} &= cy + Fsin(\omega t) \\ \dot{y} &= -x - \mu(x^2 - g(x))y \end{aligned} \qquad (3)$$

where,

$$g(x) = bx + 0.5(a - b)[|x + 1| - |x - 1|] \qquad (4)$$

Eqs (3) and (4) presents the dynamics, which are determined by the parameters $a$, $b$, $c$, $\mu$, $\omega$ and $F$.

Lyapunov exponents are used to compute the temporal separation between two adjacent orbits in the phase space under favorable conditions [42]. An $n$-dimensional system contains $n$ Lyapunov exponents. The presence of at least one positive Lyapunov exponent in the system indicated chaos. If the maximum Lyapunov exponent is negative, the orbits will converge in time, and the system will be insensitive to these conditions. If it is positive, the system has a sensitive initial condition dependency, and the distance between adjacent orbits increases exponentially; thus, it is chaotic. The Lyapunov exponent of a dynamical system can detect chaos and assess the stability of the system. Tables 1 and 2 show the Lyapunov exponents of circuit (1) for various sampling times, observation times, and initial values obtained using the Wolf algorithm [42].

In this study, the parameter values $a = -1.25$, $b = -0.68$, $c = 5$, $\mu = \frac{2}{3}$, $\omega = 0.6$, and for the parameter value $F = 3.25$, $F = 3.5$ observation periods$(T) = 100$, sampling time $\Delta t = 0.5$,. The circuit demonstrates the $(+, -)$ characteristics of the LEs under varying conditions, sample intervals, and observation times.

The LEs of system (1) by the Wolf algorithm are shown in Table 1 with varied sample times of $\Delta t = 0.5$, and observation periods of $(T) = 100$.

Fig 2a–2c show the Lyapunov exponent for the system (1) with factors $F = 3.25$ and the initial conditions $(x, y) = (0.1, 0.1)$, $(x, y) = (0.1, 0.5)$, and$(x, y) = (0.5, 0.1)$.

**Table 1. Using the Wolf approach, LEs of system (1) for observation periods $(T) = 100$, sampling time $\Delta t = 0.5$, $a = -1.25$, $b = -0.68$, $c = 5$, $\mu = \frac{2}{3}$, and $\omega = 0.6$ are calculated.**

| Sl. No. | Parameter | Initial condition | LEs $\lambda_1, \lambda_2$ | Sign of the LEs | Nature |
|---|---|---|---|---|---|
| 1 | 3.25 | 0.1, 0.1 | 0.007959, −0.896951 | +, − | Chaotic |
| | | 0.1, 0.5 | 0.013632, −0.963990 | +, − | Chaotic |
| | | 0.5, 0.5 | 0.003202, −0.890622 | +, − | Chaotic |
| 2 | 3.5 | 0.01, 0.01 | 0.045151, −1.519011 | +, − | Chaotic |
| | | 0.05, 0.01 | 0.016661, −1.325427 | +, − | Chaotic |
| | | 0.01, 0.05 | 0.045087, −1.528485 | +, − | Chaotic |
| | | 0.1, 0.1 | 0.044551, −1.908953 | +, − | Chaotic |
| | | 0.5, 0.1 | 0.008278, −0.910619 | +, − | Chaotic |
| | | 0.1, 0.5 | 0.087503, −1.859982 | +, − | Chaotic |

**Table 2. Using the Wolf approach, LEs of system (1) for observation periods ($T$) = 100, sampling time $\Delta t$ = 0.5, $a$ = −1.25, $b$ = −0.68, $F$ = 3.28, $\mu = \frac{2}{3}$, $\omega$ = 0.6 are calculated.**

| Sl. No. | Parameter | Initial condition | LEs $\lambda_1$, $\lambda_2$ | Sign of the LEs | Nature |
|---------|-----------|-------------------|------------------------------|-----------------|--------|
| 1 | 4.8 | 0.1, 0.1 | 0.008278, −0.910619 | +, − | Chaotic |
| | | 0.5, 0.1 | 0.044551, −1.908953 | +, − | Chaotic |
| | | 0.1, 0.5 | 0.087503, −1.859982 | +, − | Chaotic |
| | | 0.01, 0.01 | 0.045151, −1.519011 | +, − | Chaotic |
| | | 0.05, 0.01 | 0.016661, −1.325427 | +, − | Chaotic |
| | | 0.01, 0.05 | 0.045087, −1.528485 | +, − | Chaotic |
| | | 0.05, 0.05 | 0.031312, −1.240761 | +, − | Chaotic |
| 2 | 5.1 | 0.01, 0.01 | 0.045151, −1.519011 | +, − | Chaotic |
| | | 0.05, 0.01 | 0.016661, −1.325427 | +, − | Chaotic |
| | | 0.01, 0.05 | 0.045087, −1.528485 | +, − | Chaotic |
| | | 0.05, 0.05 | 0.031312, −1.240761 | +, − | Chaotic |

Fig 3a–3f show the Lyapunov exponent for the system (1) with factors $F$ = 3.50 and the initial conditions $(x, y)$ = (0.1, 0.1), $(x, y)$ = (0.1, 0.5), $(x, y)$ = (0.5, 0.5), $(x, y)$ = (0.01, 0.01), $(x, y)$ = (0.05, 0.01), $(x, y)$ = (0.01, 0.05).

The parameter values $a = -1.25$, $b = -0.68$, $F = 3.28$, $\mu = \frac{2}{3}$, $\omega = 0.6$. For the parameter values $c$ = 4.8, $c$ = 5.1 observation periods($T$) = 100, sampling time $\Delta t$ = 0.5,. The circuit demonstrated a (+, −) characteristics of the LEs under varying conditions, sample intervals, and observation times.

The LEs of system (1) by the Wolf algorithm are shown in Table 2 with varied sample times of $\Delta t$ = 0.5 and observation periods of ($T$) = 100.

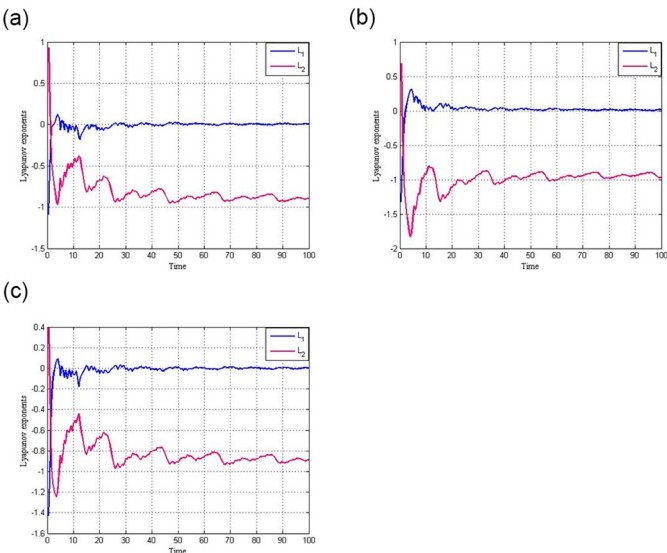

**Fig 2. Lyapunov exponent for the circuit (1) with $F$ = 3.25.** (**a**) Lyapunov exponent for circuit (1) with $F$ = 3.25 and initial condition $(x, y)$ = (0.1, 0.1) (**b**) Lyapunov exponent for circuit (1) with $F$ = 3.25 and initial condition $(x, y)$ = (0.1, 0.5) (**c**) Lyapunov exponent for circuit (1) with $F$ = 3.25 and initial condition $(x, y)$ = (0.5, 0.1).

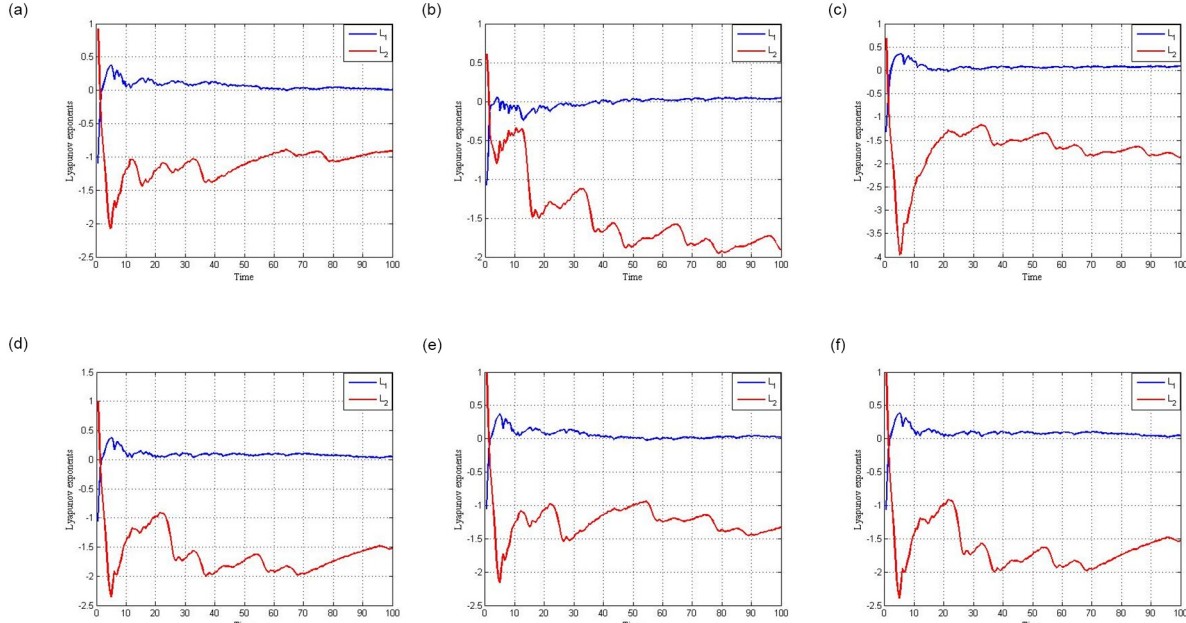

**Fig 3. The Lyapunov exponent for the circuit (1) with $F = 3.50$.** (**a**) Lyapunov exponent for circuit (1) with $F = 3.50$ and initial condition $(x, y) = (0.1, 0.1)$ (**b**) Lyapunov exponent for circuit (1) with $F = 3.50$ and initial condition $(x, y) = (0.5, 0.1)$ (**c**) Lyapunov exponent for circuit (1) with $F = 3.50$ and initial condition $(x, y) = (0.5, 0.5)$ (**d**) Lyapunov exponent for circuit (1) with $F = 3.50$ and initial condition $(x, y) = (0.01, 0.01)$ (**e**) Lyapunov exponent for circuit (1) with $F = 3.50$ and initial condition $(x, y) = (0.05, 0.01)$ (**f**) Lyapunov exponent for circuit (1) with $F = 3.50$ and initial condition $(x, y) = (0.01, 0.05)$.

The total of the Lyapunov exponents of the system (1) is

$$\lambda_1 + \lambda_2 < 0. \tag{5}$$

Hence, system (1) is a dissipative chaotic system with a chaotic attractor.

The Kaplan-Yorke fractal dimension of the new chaotic system (1) is calculated as by (6)

$$D_{KY} = 1 + \frac{\lambda_1}{|\lambda_2|} \tag{6}$$

Consequently, the new chaotic system (1) with line equilibrium becomes more complex. This view allows for the simultaneous comparison of periodic and chaotic behavior using Lyapunov exponents, phase portraits, power spectra, time series, and Poincare maps to examine the different dynamics in both local and global settings. From the investigation, there is only one long–term motion in circuit (1); for some values of the parameters, a small variation occurs, and two or more motions can form; for a few values, the circuit retains its stability owing to conditions. The term *bifurcation* refers to the phenomenon in differential equations, in which the number of solutions changes based on parameter variation [43]. If the number of solutions to differential equations changes owing to the parameter values $F$, $c$, and $\mu$. According to suitable first order differential Eq (3), the nature of the solutions of circuit(1) may abruptly change, a phenomenon known as *bifurcations* [43]. Usually, one type of motion reduces stability at a critical parameter value as it differs gradually, resulting in a new type of stable motion. The parameter values that occur at bifurcations are known as the points of bifurcation or bifurcation values [43]. This leads to circuit (1) becoming chaotic.

For circuit (1), bifurcations can be easily detected by examining variables $F$, $c$ and $\mu$.

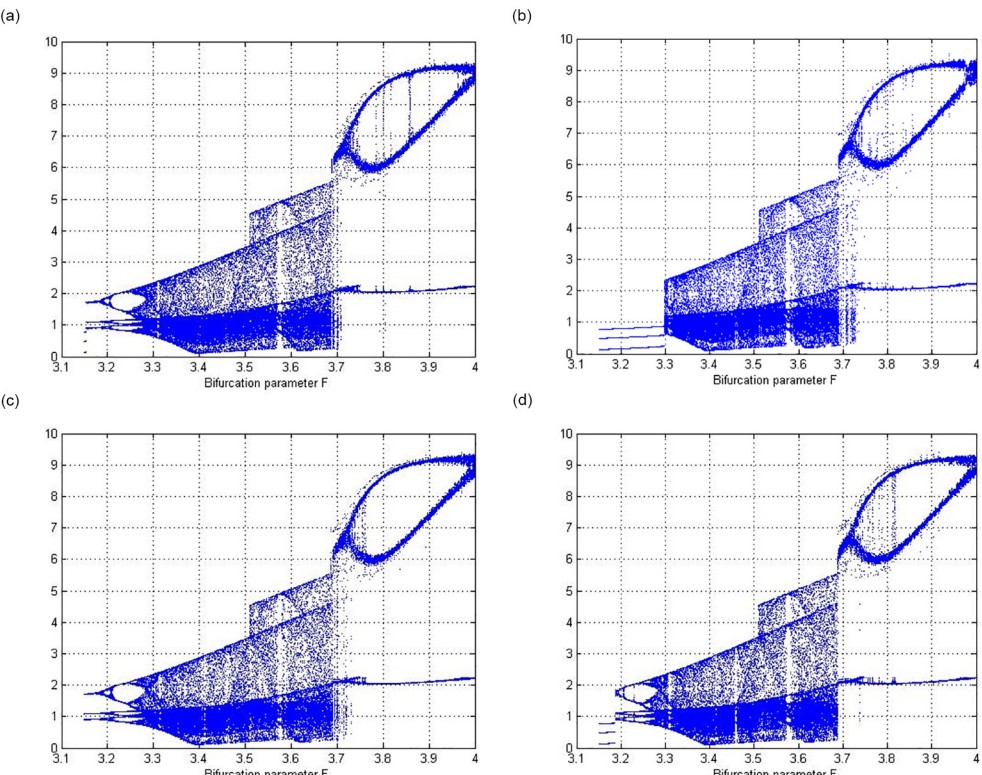

**Fig 4. Bifurcation diagram for parameter F.** (**a**) Bifurcation of parameter $F$ with initial condition $(x, y) = (0.1, 0.1)$ (**b**) Bifurcation of parameter $F$ with initial condition $(x, y) = (0.5, 0.1)$ (**c**) Bifurcation of parameter $F$ with initial condition $(x, y) = (0.1, 0.5)$ (**d**) Bifurcation of parameter $F$ with initial condition $(x, y) = (0.5, 0.5)$.

When the parameter $F$ lies between 3 and 4, circuit (1) exhibits a bifurcation.

Fig 4a to 4d show the bifurcation nature of the parameter $F$ with the conditions $(x, y) = (0.1, 0.1)$, $(x, y) = (0.5, 0.1)$, $(x, y) = (0.1, 0.5)$, and $(x, y) = (0.5, 0.5)$.

When the parameter $c$ lies between 4 and 5.1, circuit (1) exhibits a bifurcation nature.

Fig 5a shows the bifurcation nature of parameter $c$ with the conditions $(x, y) = (0.5, 0.5)$.

When parameter $c$ lies between 10 and 11, circuit (1) exhibits a bifurcation.

Fig 5b shows the bifurcation nature for parameter $c$ with the conditions $(x, y) = (0.5, 0.5)$.

When parameter $c$ lies between 1 and 2, circuit (1) exhibits a bifurcation.

Fig 5c shows the bifurcation nature for parameter $c$ with the conditions $(x, y) = (0.5, 0.5)$.

When the parameter $\mu$ lies between 1 and 2, circuit (1) exhibits a bifurcation.

Fig 5d shows the bifurcation nature for the parameter $c$ with the conditions $(x, y) = (0.1, 0.1)$.

Fig 5e shows bifurcation for the parameter $F$ with the initial condition $(x, y) = (0.1, 0.5)$ and the largest Lyapunov exponent.

Fig 5f shows the standard Lyapunov stability plot for system (1), where the two control parameters F and b vary simultaneously. Red symbolizes low chaos, and yellow symbolises higher chaos.

The Poincaré map was used to quantify the chaotic nature. The fundamental change is that an $(n - 1)^{th}$ order map replaces an $n^{th}$- order continuous time system. It was constructed using a stroboscopic device. The goal is to simplify complex systems so that they can easily attain

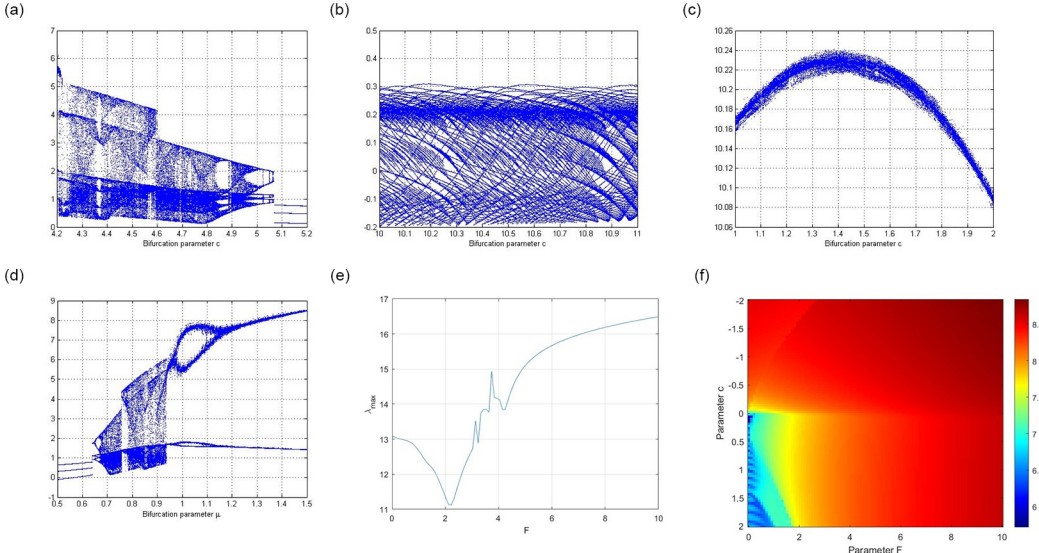

**Fig 5. Bifurcation diagram for the parameters *c* and *μ*.** (**a**) Bifurcation of parameter *c* with initial condition $(x, y) = (0.5, 0.5)$ (**b**) Bifurcation of parameter *c* with initial condition $(x, y) = (0.5, 0.5)$ (**c**) Bifurcation for parameter *c* with the initial condition $(x, y) = (0.1, 0.1)$ (**d**) Bifurcation for parameter *μ* with initial condition $(x, y) = (0.1, 0.1)$ (**e**) Bifurcation for parameter *F* with the initial condition $(x, y) = (0.1, 0.5)$ and the largest Lyapunov exponent (**f**) Standard Lyapunov stability plot for system (1) where two control parameters F and b vary simultaneously. Red symbolizes low chaos and yellow symbolizes higher chaos.

stability. In the Poincare map, periodic behavior is a fixed point [43]. In Poincare maps, closed curves or persists exhibit quasi-periodic behavior, and a distinctive group of points indicates chaos. In reality, a Poincare surface with dimensions $n - 1$ divides $R^n$ into two regions. If the Poincaré surface is suitable, the track being observed repeatedly passes through the surface. A Poincaré map is composed of this collection of crossing points. Fig 6a to 6f show the Poincare map of the chaotic behavior of circuit (1).

The power spectral density (PSD) of a chaotic system reveals its frequency content and energy distribution across different frequencies. In chaotic systems, PSD often exhibits a broad and continuous spectrum, characterized by irregular fluctuations and the absence of distinct peak frequencies. This continuous and dense spectrum signifies the intricate and non-periodic nature of chaotic dynamics, reflecting the system's unpredictable behavior and the lack of specific dominant frequencies.

Owing to their wideband nature, chaotic signals can be easily separated from periodic signals based on their frequency spectra. Fig 6f shows the power spectrum of the chaotic behavior of circuit (1). When a system exhibits chaotic behaviour, its power spectrum is described in terms of oscillations spanning a range of frequencies.

The Poincaré map acts as a simplified representation of a system's behavior within a specified region or surface of its state space. It extracts key information from the phase trajectory of the system by identifying specific points where the trajectory crosses the designated Poincaré section. This method offers a more focused perspective on system dynamics, concentrated on these selected points to provide a clearer and more concise understanding of its behavior.

A two-dimensional projection of the phase space is known as a phase portrait. This represented the instantaneous state of each state variable. The phase-portrait point represents a fixed-point solution. Closed and distinct curves in the phase portrait represent periodic and chaotic solutions, respectively.

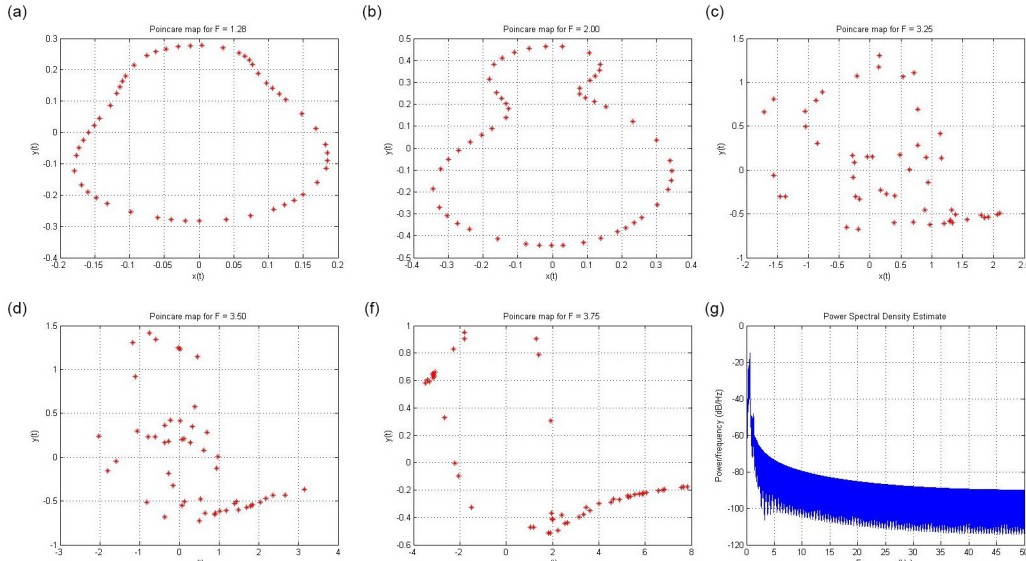

**Fig 6. Poincaré map for the parameter _F_.** (**a**) Poincaré map for parameter _F_ = 1.28 with initial condition $(x, y) = (0.1, 0.1)$ (**b**) Poincaré map for parameter _F_ = 2.0 with initial condition $(x, y) = (0.1, 0.1)$ (**c**) Poincaré map for parameter _F_ = 3.25 with initial condition $(x, y) = (0.1, 0.1)$ (**d**) Poincaré map for parameter _F_ = 3.5 with initial condition $(x, y) = (0.1, 0.1)$ (**e**) Poincaré map for parameter _F_ = 3.75 with initial condition $(x, y) = (0.1, 0.1)$ (**f**) Power Spectral Density _F_ = 3.75 with the initial condition $(x, y) = (0.1, 0.1)$.

Fig 7a–7f show the chaotic behavior of circuit (1).

Fig 8 shows an exploration of the inherent chaos in the proposed oscillator. It meticulously depicts the behavior of the system under specific parameter values $a = 1.4$, $b = 0.3$, $c = 3.1$, $\mu = 1.0$, $\omega = 1.0$, and $w = 1.0$.

Fig 8a and 8c intricately show the erratic trajectories of variables $x$ and $y$ as parameter $F$ varies between 0.2 and 0.25. These representations clearly illustrate the system's inherent unpredictability and chaotic nature.

Expanding into a three-dimensional space, Fig 8b and 8d offer captivating visual narratives of dynamic chaos that evolve over time. These projections effectively capture the complex interplay between system variables.

Collectively, Fig 8 stands as a poignant testament to the inherent chaos within the oscillator, providing profound insights into its intricate dynamics and potential cross-disciplinary application.

Fig 9 depicts the phase portrait of variables $x$ and $y$, interwoven with the local Lyapunov exponent and contour intersections. Fig 9a and 9b exquisitely reveals the phase portrait with parameter values $a = 1.4$, $b = 0.3$, $c = 3.1$, $\mu = 1.0$, $\omega = 1.0$, and $w = 1.0$. In striking contrast, Fig 9c and 9d captivatingly portray the phase portrait with parameter values $a = -1.25$, $b = -0.68$, $c = 3.1$, $\mu = (2/3)$, $\omega = 1.0$, $F = 0.2$, and $w = 1.0$.

In Fig 9a and 9c, blue represents lower values of the local Lyapunov exponent, green represents intermediate values, yellow represents moderately high values, and red represents higher values of the local Lyapunov exponent. The local Lyapunov exponent is a measure of how quickly nearby trajectories diverge in a chaotic system. Higher values of the exponent indicate greater sensitivity to the initial conditions and, therefore, more chaotic behavior.

In Fig 9a and 9c, the red-yellow regions are concentrated in the center of the phase portrait, where the trajectories are most tightly packed, and the yellow region spreads out the maximum

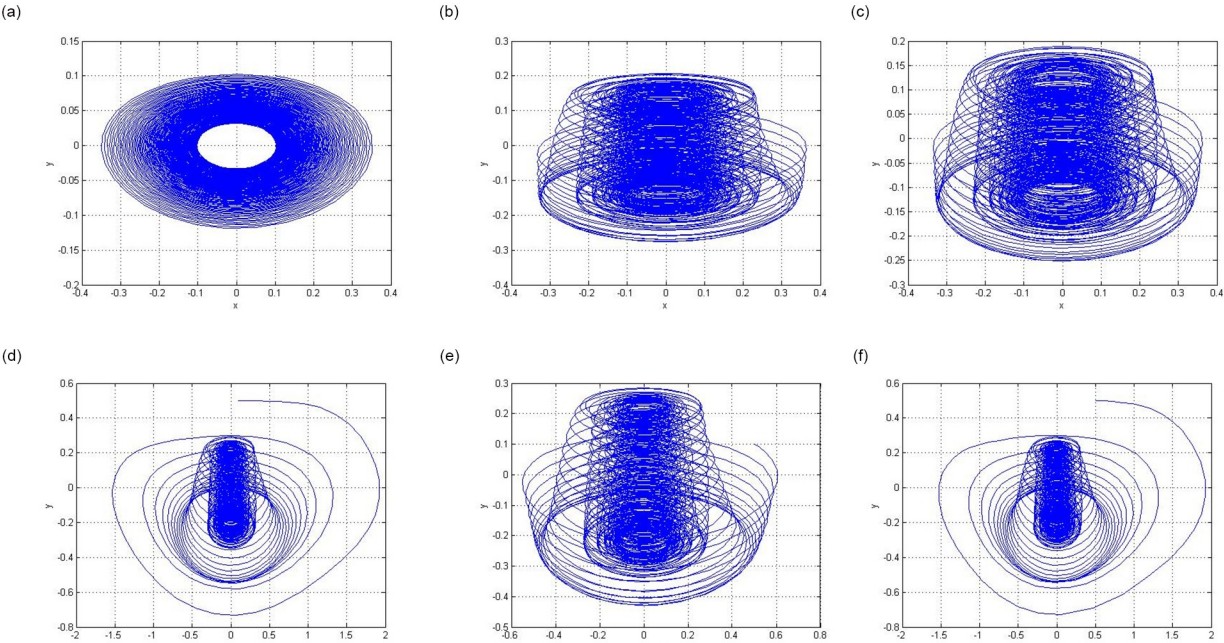

**Fig 7. Phase portrait for the chaotic attractors.** (**a**) Phase Portrait for parameter $F = 0$ with initial condition $(x, y) = (0.1, 0.1)$ (**b**) Phase Portrait for the parameter $F = 1.75$ with initial condition $(x, y) = (0.1, 0.1)$ (**c**) Phase Portrait for the parameter $F = 1.78$ with initial condition $(x, y) = (0.1, 0.1)$ (**d**) Phase Portrait for the parameter $F = 2.28$ with initial condition $(x, y) = (0.1, 0.5)$ (**e**) Phase Portrait for the parameter $F = 2.28$ with initial condition $(x, y) = (0.5, 0.1)$ (**f**) Phase Portrait for the parameter $F = 2.28$ with initial condition $(x, y) = (0.5, 0.5)$.

area. This suggests that the system was the most chaotic in this region. The blue and green regions are found in a small portion of the portrait, where the trajectories are less spread. This suggests that the system was less chaotic in these regions.

Fig 9b and 9d illustrate the spatial distribution of the local Lyapunov exponents using a color-coded contour plot. The contour lines, drawn in black, enhanced the visibility of the patterns. The blue and green shades represent regions with lower local Lyapunov exponent values. These areas are characterized by a slower divergence of nearby trajectories, indicating less chaotic behavior. The yellow and red shades represent regions with higher values of the local Lyapunov exponent. These areas are associated with a faster divergence of nearby trajectories, indicating a more chaotic behavior.

Black contour lines delineate the boundaries of distinct regions characterized by different Lyapunov exponent values. This provides a clear visual representation of the distribution of chaoticity across the phase space.

The color gradient smoothly transitions from blue (lower values) to yellow and then to red (higher values), with intermediate shades such as green. This color scheme effectively conveys varying Lyapunov exponent values across the $x - y$ phase space.

Fig 10a–10d shows the intricate dynamics of the proposed chaotic circuit, revealing its remarkable ability to transition between an ordered periodicity and intricate chaos. The figure clearly shows the periodic traits of the system, accentuating the presence of discernible patterns amidst its convoluted oscillations. Simultaneously, it encapsulates the system's chaotic essence, emphasizing its erratic nature and susceptibility to initial conditions, which are hallmark characteristics of chaotic dynamics.

Furthermore, Fig 10–10d shows the resonant behavior of the proposed chaotic circuit, a phenomenon arising from the interplay between the nonlinearities of the proposed chaotic

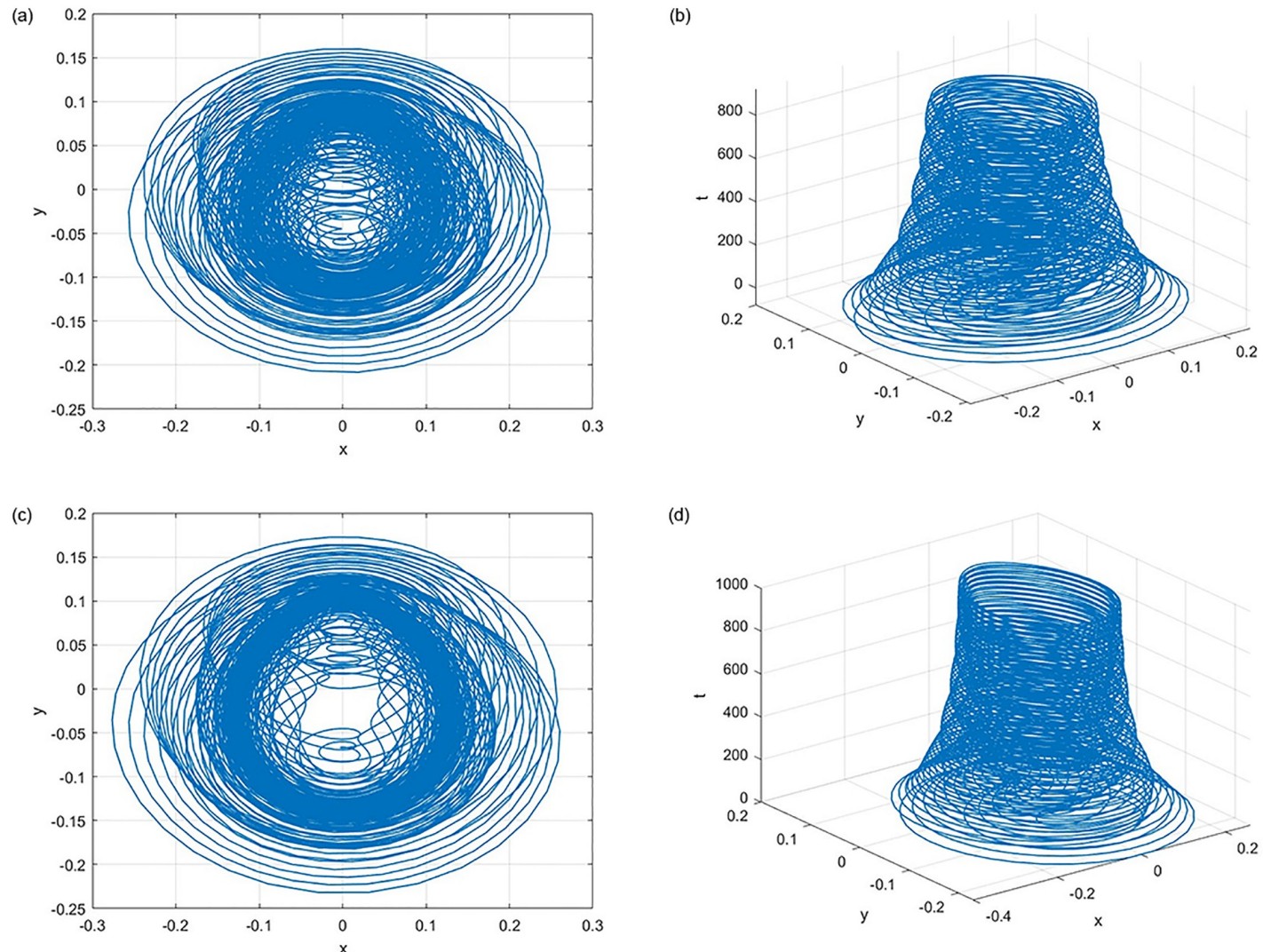

**Fig 8. Phase portrait for the chaotic attractors.** (**a**) Phase Portrait for the parameters $F = 0.2$ with initial condition $(x, y) = (0.1, 0.1)$ (**b**) Phase Portrait for parameter $F = 0.2$ with initial condition $(x, y) = (0.1, 0.1)$ (**c**) Phase Portrait for parameter $F = 0.25$ with initial condition $(x, y) = (0.1, 0.1)$ (**d**) Phase Portrait for parameter $F = 0.25$ with initial condition $(x, y) = (0.1, 0.5)$.

system. The figure demonstrates how the system response to external stimuli is amplified at specific frequencies, highlighting its potential applications in frequency filtering and signal processing.

This delicate interplay between ordered periodicity, chaotic unpredictability, and resonant behavior underscores the circuit's versatility and potential applications across diverse domains, ranging from secure communication systems to chaos-based cryptography.

In Fig 10c, the red lines indicate the potential for periodic behavior within the proposed chaotic system. This observation suggests that the system may demonstrate periodic tendencies under specific initial conditions.

Fig 10d delves into the intricate resonant behavior exhibited by the proposed chaotic circuit, shedding light on how the system responds to external stimuli in relation to parameter $F$. The figure meticulously explores the resonances of the variables $x$ and $y$ across various $F$ values

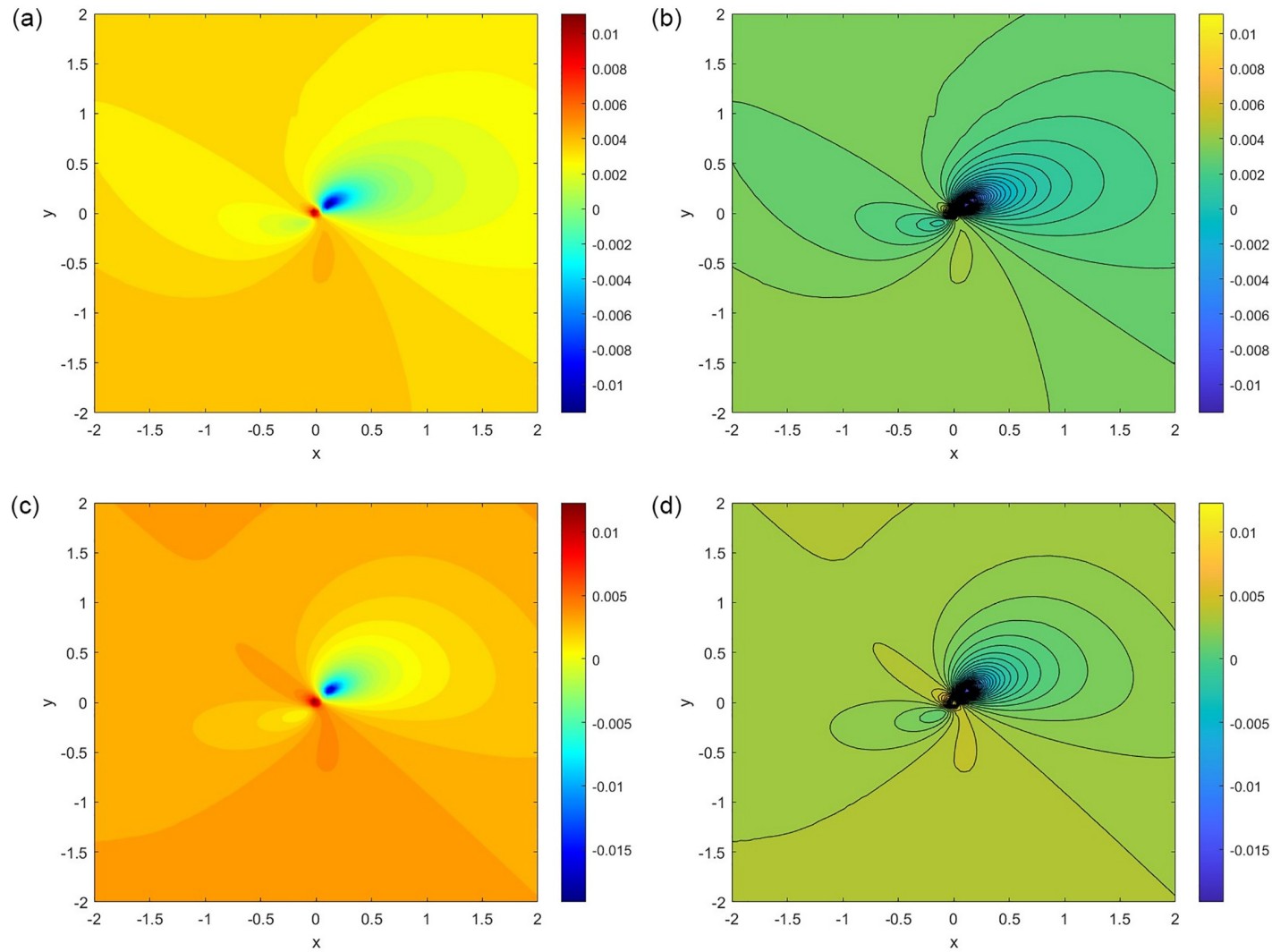

**Fig 9. Phase portrait with local Lyapunov exponent.** (**a**) Phase Portrait for the parameter $a = 1.4$, $b = 0.3$, $c = 3.1$, $\mu = 1.0$, $\omega = 1.0$, $w = 1.0$, and $F = 0.2$ with initial condition $(x, y) = (0.1, 0.1)$ (**b**) Phase Portrait for the parameter $a = 1.4$, $b = 0.3$, $c = 3.1$, $\mu = 1.0$, $\omega = 1.0$, $w = 1.0$, and $F = 0.2$ with initial condition $(x, y) = (0.1, 0.1)$ (**c**) Phase Portrait for the parameter $a = -1.25$, $b = -0.68$, $c = 3.1$, $\mu = \frac{2}{3}$, $\omega = 1.0$, $F = 0.2$, and $w = 1.0$ with initial condition $(x, y) = (0.1, 0.1)$ (**d**) Phase Portrait for the parameter $a = -1.25$, $b = -0.68$, $c = 3.1$, $\mu = \frac{2}{3}$, $\omega = 1.0$, $F = 0.2$, and $w = 1.0$ with initial condition $(x, y) = (0.1, 0.1)$.

(0.1, 0.5, 1, and 1.5), unveiling the dynamic nature of the system. It vividly portrays the shifting resonance peaks for both variables, showcasing the system's sensitivity to changes in F and suggest F as an effective tool for tuning the response to external signals.

In addition, the figures highlights the modulation of the resonance peak amplitudes with varying F. This modulation implies that F can regulate the magnitude of the system at the resonant frequencies.

These observations emphasize the profound influence of F on shaping the resonant behavior of chaotic circuits. This intricate relationship between F and the system resonances allows researchers to customize the circuit dynamics for specific applications.

In essence, Fig 10d offers valuable insights into the resonance characteristics of the chaotic circuit, illustrating their reliance on the parameter F. These insights pave the way for designing and optimizing circuits that exhibit tunable resonant behavior for diverse applications.

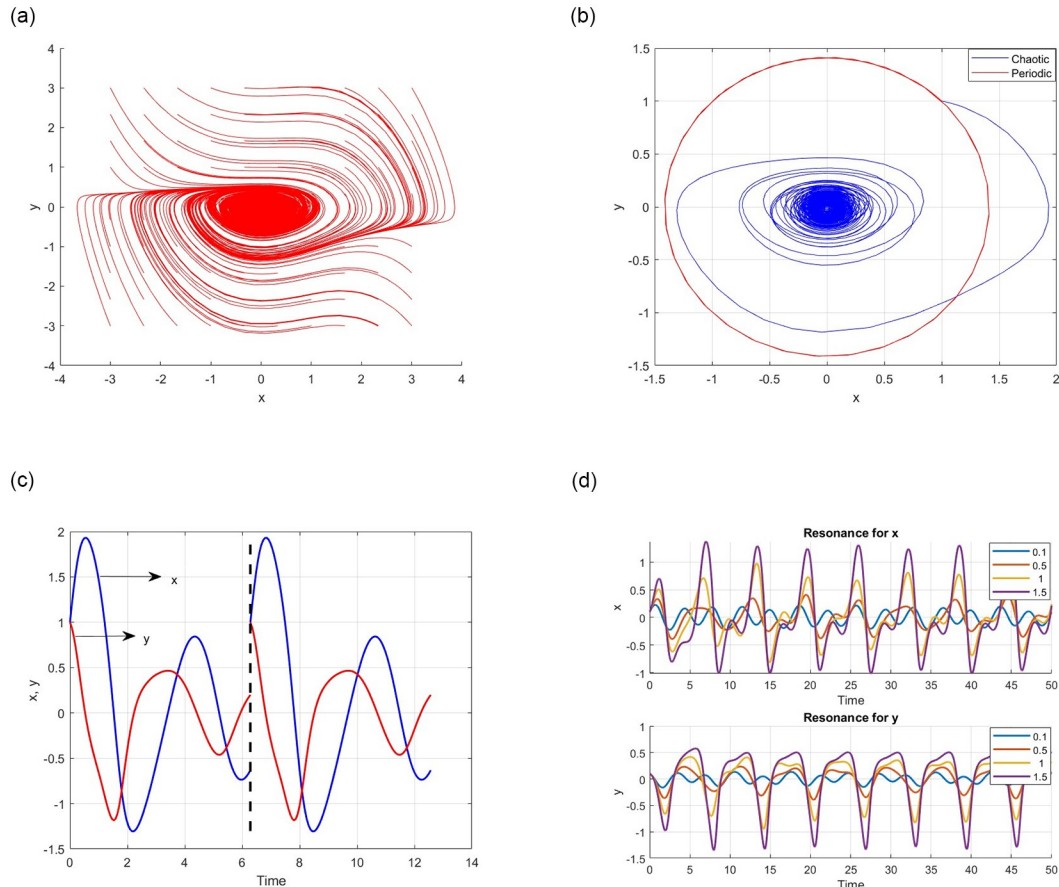

**Fig 10. Portrait for the periodic, chaotic and resonance nature of the proposed system.** (**a**) Phase Portrait for the parameter $F = 0.2$ with initial condition $(x, y) = (0.1, 0.1)$ (**b**) Phase Portrait for the parameter $F = 0.2$ with initial condition $(x, y) = (0.1, 0.1)$ (**c**) Phase Portrait for the parameter $F = 0.25$ with initial condition $(x, y) = (0.1, 0.1)$ (**d**) Phase Portrait for the parameter $F = 0.25$ with initial condition $(x, y) = (0.1, 0.5)$.

When addressing a designed chaotic system, the most significant topics of discussion are the synchronization and control of the chaotic system. The subsequent section of this article focuses on managing the chaotic circuit by applying a backsteeping control.

## Backstepping control for the chaotic circuit

Backstepping control is a nonlinear approach widely used in managing intricate systems with uncertain dynamics. Its primary objective is to stabilize and regulate systems that traditional linear control methods may struggle to handle effectively. This method is particularly significant when dealing with nonlinear systems, such as the Chua system, which is characterized by chaotic and intricate unpredictable behavior dynamics.

The essence of backstepping lies in crafting nonlinear controllers that facilitate the stabilization and control of systems entangled in complex nonlinear dynamics. Its strength emerges from its ability to counter uncertainties and disruptions within systems, making it a suitable choice for systems riddled with unknown or fluctuating parameters. Moreover, backstepping excels in ensuring precise trajectory tracking, even amidst disturbances, by aligning the path of the system with the desired trajectories.

Specifically applied to the Chua system, notorious for its chaotic and nonlinear traits, back-stepping control aims to curtail or stabilize this chaotic behavior. This enables the redirection of the system's state toward predefined trajectories or specific operational points, which are crucial in applications requiring predetermined behaviors or patterns. Moreover, this control method bolsters the system's resilience against external disturbances, enhancing its ability to navigate the uncertainties or noise inherent in Chua systems.

A suitable Lyapunov function and feedback controller are looped using the backstepping technique. This stabilizes chaotic systems with strict global feedback. In this section, the proposed chaotic circuit is examined using the backstepping method.

The state-space representation of the chaotic circuit (1) with backstepping control is defined as follows:

$$
\begin{aligned}
\dot{x} &= cy + F sin(\omega t) \\
\dot{y} &= -x - \mu(x^2 - g(x))y + u.
\end{aligned}
\tag{7}
$$

where,

$$
g(x) = bx + 0.5(a - b)[|x + 1| - |x - 1|]
\tag{8}
$$

In (7) $x$ and $y$ are state variables and $u$ is the controller.

First consider the first state of the defined Eq(7).

$$
\dot{x} = cy + F sin(\omega t).
\tag{9}
$$

$y$ is considered to be a virtual controller in this scenario.

The appropriate Lyapunov function, which can stabilise the system, is defined as

$$
V_1 = \frac{1}{2}x^2.
\tag{10}
$$

Differentiate Eq(10) with regards to t,

$$
\dot{V}_1 = x(cy + F sin(\omega t)).
\tag{11}
$$

Let $y$ be the virtual controller and it is defined as

$$
y = \alpha(x, \ t) \text{ and } \alpha(x, \ t) = -\frac{(F sin(\omega t) + x)}{c}.
\tag{12}
$$

With regard to the implication of substituting (12) for (11),

$$
\dot{V}_1 = -x^2
\tag{13}
$$

This proves to be a negative definite function. Therefore, the system is globally asymptotically stable, according to Lyapunov stability theory.

Now the relation between $y$ and $\alpha$ is defined by

$$
\beta = y - \alpha.
\tag{14}
$$

Consider the subsystem $(x, \beta)$

$$\dot{x} = c\beta - x$$

$$\dot{\beta} = -x - \mu(x^2 - g(x))(\alpha + \beta) + \frac{F\omega cos(\omega t)}{c}$$

$$+ (\alpha + \beta) + \frac{Fsin(\omega t)}{c} + u.$$

(15)

where $x$ is the state variable. Now consider the Lyapunov function

$$V_2 = V_1 + \frac{1}{2}\beta^2.$$

(16)

The derivative of $V_2$ with respect to $t$ is calculated as

$$\dot{V}_2 = -x^2 + \beta(cx - x - \mu(x^2 - g(x))(\alpha + \beta)$$

$$+ \frac{F\omega cos(\omega t)}{c} + (\alpha + \beta) + \frac{Fsin(\omega t)}{c} + u).$$

(17)

The backstepping control $u$ has been defined as

$$cu = cx(1 - x) + c(\mu(x^2 - g(x)) + 1)(\alpha + \beta)$$

$$- F(\omega cos(\omega t) - sin(\omega t)) - c\beta.$$

(18)

Subsituting (18) for (17), we obtain

$$\dot{V}_2 = -x^2 - \beta^2,$$

(19)

which performs a defined negative function. According to the Lyapunov stability theory [44], chaotic circuit (1) is globally asymptotically stable.

For the numerical simulation, the conditions of system (7) were taken as $(x(0), y(0)) = (0.5, 0.5)$. Fig 11 shows the complete stabilization of the new chaotic system with line equilibrium.

Integrating chaotic behavior into cryptography protocols is one of the most interesting applications of chaotic systems. The developed chaotic oscillator can be utilized for the encryption and decryption of both text and images. The remainder of this paper is organized as follows:

## Data encryption

The rapid development of internet technology has made data security important. Information is typically represented by digital images. Digital images can be secured using image encryption methods [26]. The purpose of text and image encryption is to change the text and image from understandable to unintelligible pattern [30, 45]. Typically, chaotic models are used to create secure image and text crypto systems [25, 46, 47]. An image and text crypto system that makes use of the useful characteristics of chaotic flow is proposed. Therefore, this section focus on the proposed text and image cryptosystem and its performance evaluation. Designing secure cryptography primitives using chaos-based methods plays an important role [48–50]. This ensures that the date cannot be predicted by the hackers. This section presents and analyzes the proposed chaos-based cryptography mechanism.

The text may be encrypted and decrypted using well-known quotations, the first of which is "*Veni, Vidi, Vici*," which is the name of the Latin phrase that is credited to Julius Caesar and

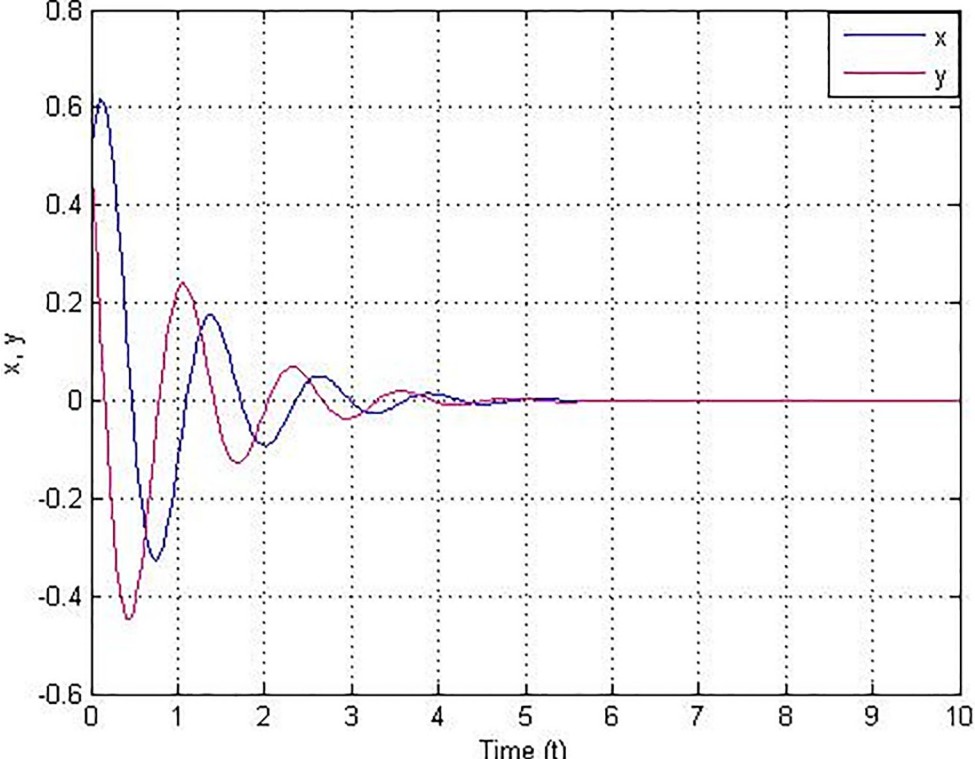

**Fig 11. Stabilisation under backstepping control.**

means "*I came, I saw, I conquered.*" The following is titled "*I Have Dream*". This is the first part of the legendary "I Have a Dream" speech that was delivered by Martin Luther King Jr.

The simulation results revealed that the decrypted texts were so noisy that no data could be extracted.

## Encryption algorithm for text

Encryption algorithms are mathematical functions that are used to convert original plain-text data into encrypted, unreadable cipher text, and vice versa. The purpose of encryption is to secure sensitive information from unauthorized access by transforming it into a format that can only be read by someone with a proper decryption key.

Chaotic oscillator-based encryption algorithms use mathematical models of chaotic systems to generate encryption keys to secure digital communications. These algorithms utilize the unpredictable and complex behavior of chaotic systems to produce encryption keys that are difficult for attackers to crack. However, the security of chaotic oscillator-based encryption algorithms is still a subject of ongoing research. The proposed text-crypto system exploits the *S-box* method. The created sequence is used to first replace the plain text, after which some data about the replaced text are collected and utilized to alter the initial conditions of the chaotic oscillator. The chaotic oscillator system is solved using the revised initial conditions to produce two sequences: one for rearranging the rows of the replaced text, and the other for rearranging its columns. Create a final text to produce the cipher text. Algorithm 1 describes the proposed text encryption technique. The security of an encryption algorithm depends on the strength of the key and algorithm itself.

**Algorithm 1**: Text Encryption Using Chaotic System

**Data:** Text Message, Chaotic System Parameters (`a`, `b`, `c`, `μ`, `ω`, `F`, `w`), Initial Conditions, S-box

**Result:** Encrypted Text

**Define parameters for the chaotic system:**

$a = 1.4, b = 8.3, c = 3.1, \mu = \frac{2}{3}, \omega = 1.0, F = 3.25, w = 1.0;$         // Adjust as needed

**Define the ODE system for the chaotic system using the parameters**

**Initialise initial conditions:** [0.1;0.5]

**Set the time span for the ODE solver:** [0, 1000]

**Generate the chaotic sequence by solving the ODE system**

**Define the text message to be encrypted:**     // Input message

**Initialize the cipher text as an array**

**Define the S-box for substitution:** $s\_box$ = randperm(256)

**for** *each character in the text message* **do**

 **Convert the character to its ASCII value:**
 *char_value* = ASCII(character)
 **Get the corresponding chaotic value from the chaotic sequence**
 **Perform S-box substitution on** *char_value* **using** *s_box*:
 *sbox_substitution = s_box[char_value + 1]*
 **Ensure that both operands are integers:**
 *encrypted_char_value = bitxor(sbox_substitution, (chaotic_value ×*
 *255))*
 **Append** *encrypted_char_value* **to the cipher text array**

**Convert the cipher text array to characters**

**Display the encrypted text**

## Decryption algorithm for text

A decryption algorithm is a process used to convert encrypted information back into its original unencrypted form. The decryption algorithm typically uses a key or password that must match the encryption key or password used to encrypt the information. The decryption algorithm reverses the encryption process and transforms the cipher-text back into the original plain text. The strength of the employed encryption algorithm and the security of the decryption key affect the security of the decryption algorithm.

The decryption algorithm of the proposed cryptosystem is the reverse of that of the encryption algorithm. In a closed setting or by using an appropriate asymmetric cryptography mechanism for key distribution, the sender and receiver may reveal the key parameters used in the encryption process beforehand. After the key has been encrypted, an asymmetric cryptography algorithms is used to carry out the decryption process on the receiver's device, sharing the key's value with both the sender and receiver. The proposed decryption algorithm is described in Algorithm 2.

**Algorithm 2**: Text Decryption Using Chaotic System

**Data:** Encrypted Text, Chaotic System Parameters (`a`, `b`, `c`, `μ`, `ω`, `F`, `w`), Initial Conditions, S-box

**Result:** Decrypted Text

**Define parameters for the chaotic system:**

$a = 1.4, b = 8.3, c = 3.1, \mu = \frac{2}{3}, \omega = 1.0, F = 3.25, w = 1.0;$         // Match the parameters used for encryption

**Define the ODE system for the chaotic system using the parameters**

**Initialize initial conditions:** [0.1;0.5]     // Use the same initial conditions as in encryption

**Set the time span for the ODE solver:** [0, 1000]     // Match the time span used for encryption

**Generate the chaotic sequence by solving the ODE system**

**Initialize the decrypted text as an array**

```
Define the S-box for substitution: s_box = inverse_sbox;      // Match
the S-box used for encryption
for each character in the encrypted text do
  Convert the character to its ASCII value:
  char_value = ASCII(character)
  Get the corresponding chaotic value from the chaotic sequence
  Perform inverse S-box substitution on char_value using s_box: sbox_
substitution = s_box⁻¹(char_value)
  Ensure that both operands are integers:
  decrypted_char_value = bitxor(sbox_substitution, (chaotic_value ×
255))
  Append decrypted_char_value to the decrypted text array
Convert the decrypted text array to characters Display the decrypted
text
```

## Performance evaluation of text cryptosystems

To ensure the efficacy of the proposed text cryptosystem, performance testing was performed using a PC preinstalled with MATLAB 2023. $a = -1.25$, $b = -0.68$, $c = 1.8$, $F = 3.28$, $\mu = \frac{2}{3}$, $\omega = 0.6$, $x_0 = 0.1$ and $y_0 = 0.1$ are the primary significant parameters. Fig 12 depicts the encryption and decryption performance of the proposed algorithm.

The effectiveness of any text encryption algorithm depends on two factors: its encryption duration and resistance to various assaults, including brute force, statistical cryptography, and key sensitivity. To demonstrate the effectiveness of the proposed text-encryption technique, these factors are described in the following subsections.

## Correlation performance

A correlation attack is a type of cryptanalysis in which an attacker attempts to determine the connections between the encrypted data and plaintext. The idea is to find patterns or relationships that can be used to deduce the original plaintext or encryption keys. Correlation coefficient analysis is a type of statistical analysis.

This analysis visually depicts the distribution of adjacent bits in both the original and encrypted texts. In general, plain text should have large correlations with neighboring bits, whereas encrypted text should have few connections with its adjacent bits.

The correlation coefficients between each bit value in plain text and its neighboring bits are very close to one, whereas they are very close to zero in cipher text. When computing the correlation coefficients for plain and encrypted text, the bits were chosen at random. Table 3 shows the correlation coefficients for plain texts and their comparable cipher equivalents, where the cipher text correlation values were quite close to 0. Fig 13 also shows the correlation

| Plain Text | Encrypted Text |
|---|---|
| "Hello World!" | ÚÔ⌷⌷È⌷Dä*ìk§ |
| "Veni, Vidi, Vici" | ⌷æ⌷¿⌷D#⌷ìÀ⌷Úä]] |
| "I Have Dream" | /ñrq<⌷Æä⌷Äá> |

**Fig 12. Experimental execution results in plain text.**

**Table 3. Correlation coefficient between plain text and encrypted text.**

| Plain and Encrypted Text | Correlation Value |
|---|---|
| From Fig 12 first plain and encrypted text | 0.3314 |
| From Fig 12 second plain and encrypted text | 0.5321 |
| From Fig 12 third plain and encrypted text | 0.6667 |

distribution for plain and encrypted text. The results revealed that the proposed cryptosystem can withstand correlation analysis.

The correlation coefficient is calculated using the formula (20) shown below.

$$r_{xy} = \frac{\sum_{i=1}^{N} \left(x_i - \frac{1}{N}\sum_{i=1}^{N} x_i\right) \sum_{i=1}^{N} \left(y_i - \frac{1}{N}\sum_{i=1}^{N} y_i\right)}{\sqrt{\sum_{i=1}^{N} \left(x_i - \frac{1}{N}\sum_{i=1}^{N} x_i\right)^2 \sum_{i=1}^{N} \left(y_i - \frac{1}{N}\sum_{i=1}^{N} y_i\right)^2}} \tag{20}$$

The intensity values of the two adjacent texts are shown here as $x$ and $y$. Fig 13a–13c displays the correlations discovered for both the unencrypted and original texts. In the original text and encrypted version, the correlations between the bits are very strong and very low, respectively. The average correlation coefficients for both plain and encrypted text are

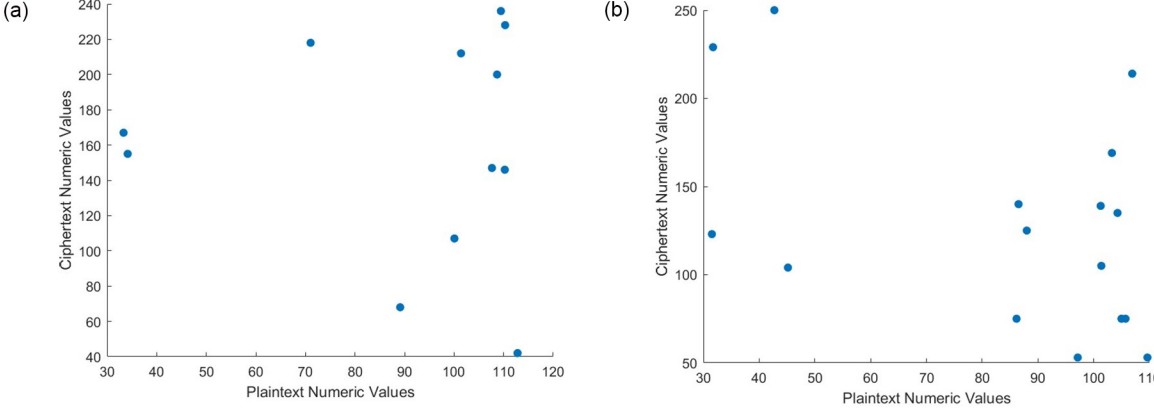

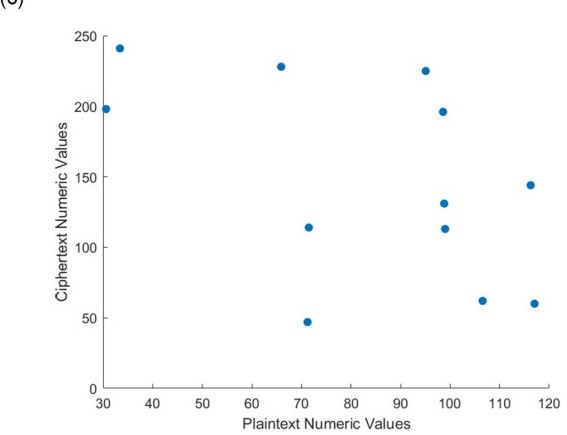

**Fig 13. Scatter Plot analysis for text data.** (**a**) Scatter Plot for Text—1 (**b**) Scatter Plot for Text—2 (**c**) Scatter Plot for Text—3.

summarized in Table 3. The proposed approach performs strong encryption in terms of the degree of correlation between adjacent bits, proving that it has good confusion and diffusion qualities.

## Histogram test for text encryption

Histogram analysis is a type of security analysis that reveals the statistical properties of an encrypted text. In a histogram test, the encrypted text is divided into unique ASCII values, and the frequency of each intensity level is counted and plotted as a histogram. The histogram of the encrypted text frequency is then compared with the expected ASCII value. This suggests that the encryption process has added sufficient randomness to the text and that the encryption algorithm works as intended. Histogram tests can be useful in detecting certain types of encryption attacks, such as substitution attacks, in which the attacker substitutes one bit value for another in the encrypted image.

A good text cryptosystem approach should ensure the consistency of histograms for different encrypted texts. In Fig 14a–14c, the histograms of the plain text are contrasted with those of their equivalent cipher text, which are consistent with one another. The histograms of the equivalent cipher text are similar to one another. The similarity of the histograms for the ciphered text is checked using a quantitative test known as variance (Var), which can be expressed as shown in Eq (21).

$$Var(T) = \frac{1}{255^2} \sum_{p=0}^{255} \sum_{q=0}^{255} \frac{(t_p - t_q)^2}{2} \tag{21}$$

where $t_p$ and $t_q$ are pixel numbers whose gray values are $p$ and $q$, respectively, and

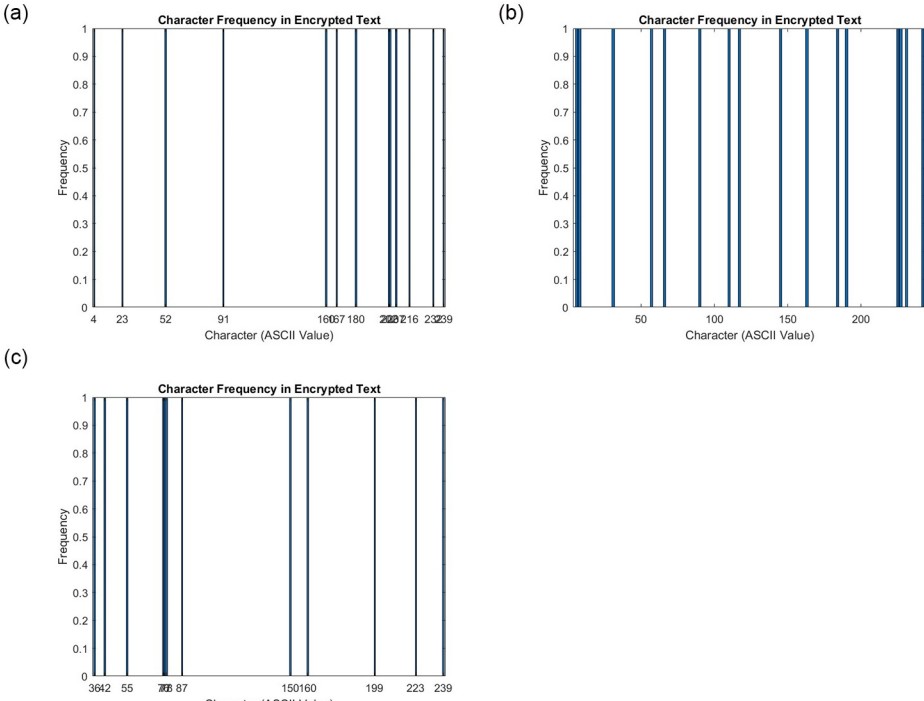

**Fig 14. Histogram analysis for text data.** (**a**) Histogram for Text—1 (**b**) Histogram for Text—2 (**c**) Histogram for Text—3.

**Table 4. Histogram variance for plain and cipher text.**

| Text | Variance Value | |
|------|------|------|
| | **Plain** | **Cipher** |
| Hello World! | 872.27 | 3731.27 |
| Veni, Vidi, Vici | 818.06 | 4591.50 |
| I Have Dream | 865.72 | 5049.90 |

$T = t_0, t_1, \cdots, t_{255}$ is the sequence of the histogram values. Table 4 displays the histogram variance results for the tested text both before and after encryption. Low numbers indicate a homogenous distribution in the histograms.

## Information entropy for text

Information entropy is a measure of the amount of uncertainty or randomness in a given dataset. In the context of text, information entropy measures how unpredictable or random characters or symbols are in the text. Text with high entropy has more unpredictable characteristics, whereas text with low entropy is more regular and predictable.

The formula (22) for calculating the information entropy of a text is

$$H(x) = -\sum_{i=1}^{N} P(x_i)log_2 P(x_i) \tag{22}$$

where $n$ is the number of unique characters or symbols in the text, $x_i$ represents each unique character or symbol in the text; $P(x_i)$ is the probability of occurrence of character $x_i$ in the text. Table 5 depicts the entropy value between plain and encrypted text.

## Key sensitivity for text encryption

Key sensitivity in the context of text encryption refers to the sensitivity of encrypted text to small changes or perturbations in the encryption key. It quantifies the extent to which a slight modification of the encryption key affects the resulting ciphertext. A lower key sensitivity indicates that small changes in the key produce minimal differences in the ciphertext, thereby making the encryption more robust to key variations.

One common metric used to calculate the key sensitivity is the Hamming distance. The Hamming distance measures the number of different bits between two binary strings, such as two ciphertexts.

A lower key sensitivity value suggests that small changes in the encryption key have a minor impact on the ciphertext, indicating a more robust encryption algorithm. Conversely, a higher key sensitivity value implies that small key variations lead to significant differences in ciphertext, which may indicate a less robust encryption scheme.

**Table 5. Information entropy for plain and cipher text.**

| | Information Entropy | |
|------|------|------|
| Text | **Plain (bits/character)** | **Cipher (bits/character)** |
| Hello World! | 3.02 | 3.58 |
| Veni, Vidi, Vici | 2.73 | 3.75 |
| I Have Dream | 3.09 | 3.59 |

**Table 6. Key sensitivity for text encryption.**

| Text | Key Sensitivity |
|------|-----------------|
| Text 1 | 0.125 |
| Text 2 | 0.102 |
| Text 3 | 0.125 |

Key sensitivity analysis is essential for evaluating the security and stability of encryption algorithms, particularly in scenarios where keys may be exposed to noise or slight alterations.

The key sensitivity is calculated as (23)

$$KS = \frac{D_H(C_0, C_1)}{L} \tag{23}$$

where $D_H(C_0, C_1)$ is the Hamming distance between $C_0$ and $C_1$, L is the total number of bits in the ciphertext(length of $C_0$ or $C_1$), $C_0$ is the original ciphertext generated using the original encryption key, and $C_1$ is the ciphertext generated using a perturbed (modified) encryption key.

The Hamming distance $D_H$ between $C_0$ and $C_1$ is calculated as (24)

$$D_H(C_0, C_1) = \sum_{i=1}^{L} |C_0(i), C_1(i)| \tag{24}$$

where $C_0(i)$ and $C_1(i)$ are $i^{th}$ bits of $C_0$ and $C_1$ respectively. Table 6 shows the key sensitivities of plain and encrypted text.

This analysis helps identify the strengths and weaknesses of each text based cryptosystem. The key benefits of the proposed encryption technique are, as follows;

1. The statistics in Table 3 show that the proposed cryptosystem can withstand correlation analysis because the correlation coefficients of the cipher text are in the range of 0 and 1.

2. The charts in Fig 14 demonstrate the consistency of the cipher text histograms are with one another.

3. The results in Table 4 demonstrate the cipher text strong histogram uniformity.

4. According to the data in Table 5, the entropy values for cipher text are approximately four, and the suggested cryptosystem can withstand entropy analysis.

5. The results in Table 6 reveal that the proposed text encryption method has high key sensitivity, meaning that even minor changes to the initial keys can significantly affect the output.

Image encryption and decryption are laborious and artistic processes. If a chaotic system is involved, the system becomes more complex. An image cryptosystem based on the designed chaotic system is covered in the future.

## Encryption algorithm for image

Encryption algorithms are mathematical functions used to convert original plain image data into encrypted, unreadable cipher images, and vice versa. The purpose of encryption is to secure sensitive information from unauthorized access by transforming it into a format that can only be read by someone with the appropriate decryption key [51].

Chaotic oscillator-based encryption algorithms use mathematical models of chaotic systems to generate encryption keys to secure digital communications. These algorithms utilize the unpredictable and complex behavior of chaotic systems to produce encryption keys that are difficult for attackers to crack. However, the security of chaotic oscillator-based encryption algorithms is still a subject of ongoing research.

The proposed image cryptography system takes advantage of the S-box method. The created sequence is first used to replace the plain image, after which some data about the replaced image are collected and utilized to alter the initial conditions of the chaotic oscillator. The chaotic oscillator system is solved using the revised initial conditions to produce two sequences: one for rearranging the rows of the replaced image and the other for rearranging its columns. Finally, an image was created to produce a cipher image. Algorithm 3 describes the proposed image-encryption technique. The security of an encryption algorithm depends on the strength of the key and algorithm itself.

**Algorithm 3**: Image Encryption Using Chaotic System

```
Data: Initial Conditions (x₀, y₀) and Control Parameters (a, b, c, μ, ω)
Result: Encrypted Image
Function GenerateChaoticSequence(x₀, y₀, a, b, c, μ, ω, R × C × N)
  Result: Chaotic Sequence
  ;        // Obtain a sequence of length R × C × N using the key
parameters
Data: [R, C, N] ← size(plain_image)
Data: [xkey, ykey] ←
      GenerateChaoticSequence(x₀, y₀, a, b, c, μ, ω, R × C × N)
Data: EncryptedImage ← zeros(R, C, N)
;        // Initialize an array to store the encrypted image
for k ← 1 to N do
  for i ← 1 to R do
    for j ← 1 to C do
      Data: pixel_value ← double(plain_image(i, j, k))
      Data: chaotic_value ← chaotic_sequence((k – 1) mod length(t)
            + 1)
      Data: sbox_substitution ← s_box(pixel_value + 1)
      Data: encrypted_pixel_value ← bitxor((sbox_substitution),
            (chaotic_value × 255))
      Data: EncryptedImage(i, j, k) ← encrypted_pixel_value
Data: Display the encrypted image
```

Fig 15 shows how the image encryption process works using a chaotic value array and S-box substitution. This method requires three parameters: the original image, initial conditions and parameter values for the chaotic system (1), and S-box. When these are inputs, the algorithm generates an encrypted image as its output.

## Decryption algorithm for image

A decryption algorithm is a process used to convert encrypted information back into its original unencrypted form. The decryption algorithm typically uses a key or password, that must match the encryption key or password used to encrypt the information. The decryption algorithm reverses the encryption process and transforms the cipherical image back into the original plain image. The strength of the employed encryption algorithm and the security of the decryption key affect the security of the decryption algorithm.

The decryption algorithm of the proposed cryptosystem is the reverse of that of the encryption algorithm. In a closed setting or by using an appropriate asymmetric cryptography mechanism for key distribution, the sender and receiver may reveal the key parameters used in the encryption process beforehand. After the key has been encrypted, an asymmetric cryptography

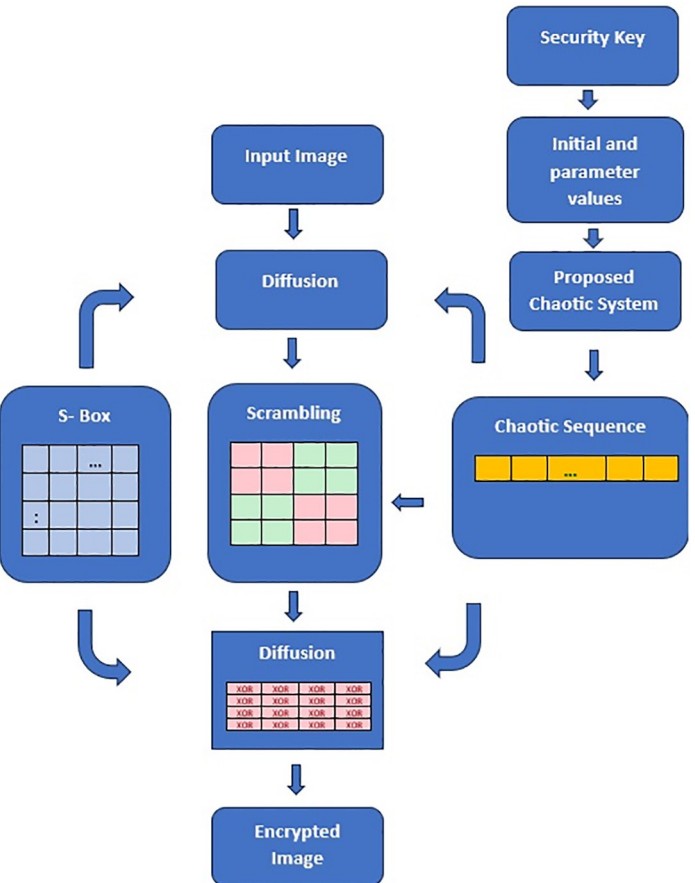

**Fig 15. Schematic encryption processes.**

algorithm is used to carry out the decryption process on the receiver's device, sharing the key's value with both the sender and receiver. The proposed decryption algorithm is described in Algorithm 4.

**Algorithm 4**: Image Decryption Using Chaotic System

```
Data: Initial Conditions (x₀, y₀) and Control Parameters (a, b, c, μ,
      ω), Encrypted Image
Result: Decrypted Image
Function GenerateChaoticSequence(x₀, y₀, a, b, c, μ, ω, R × C × N)
  Result: Chaotic Sequence
  ;        // Obtain a sequence of length R × C × N using the key
parameters
Data: [R, C, N] ← size(encrypted_image)
Data: [xkey, ykey] ←
      GenerateChaoticSequence(x₀, y₀, a, b, c, μ, ω, R × C × N)
Data: DecryptedImage ← zeros(R, C, N)
;        // Initialize an array to store the decrypted image
for k ← 1 to N do
  for i ← 1 to R do
    for j ← 1 to C do
      Data: encrypted_pixel_value ← double(encrypted_image(i, j, k))
      Data: chaotic_value ← chaotic_sequence((k - 1) mod length(t)
            + 1)
```

```
      Data: sbox_substitution ← s_box⁻¹(encrypted_pixel_value)
      Data: decrypted_pixel_value ← bitxor((sbox_substitution),
            (chaotic_value × 255))
      Data: DecryptedImage(i, j, k) ← decrypted_pixel_value
Data: Display the decrypted image
```

## Performance evaluation of image cryptosystems

To ensure the efficacy of the proposed image cryptosystem, performance tests were performed using a PC preinstalled in MATLAB 2023. "Aeroplane," "baboon," "boat," "house," and "pepper" are the five standard images in the dataset of used images. The most significant parameters are $a = -1.25$, $b = -0.68$, $c = 1.8$, $F = 3.28$, $\mu = \frac{2}{3}$, $\omega = 0.6$, $x_0 = 0.1$, and $y_0 = 0.1$. Fig 16a–16j depicts the suggested algorithm's encryption and decryption performance.

The effectiveness of any image encryption algorithm depends on two factors: its encryption duration and its resistance to various assaults, including brute force, statistical cryptanalysis, and differential cryptanalysis. To demonstrate the effectiveness of the proposed image-encryption technique, these factors are described in the following subsections.

## Correlation performance

A correlation attack is a type of cryptanalysis in which an attacker attempts to discern the connections between encrypted data and plaintext. The objective is to identify the patterns or relationships that can be exploited to deduce the original plaintext or encryption key. A different form of statistical analysis was correlation coefficient analysis.

This analysis visually illustrates the distribution of adjacent pixels in both the original and encrypted images. In general, plain images should exhibit large correlations with neighboring pixels, whereas encrypted images should have few connections with adjacent pixels.

The correlation coefficients between each pixel value in a plain image and its neighboring pixels are very close to one, whereas they are very close to zero in cipher images. When computing the correlation coefficients for plain and encrypted images, the pixels were chosen randomly. Table 7 shows the correlation coefficients for plain images and their comparable cipher equivalents, where the cipher image correlation values were close to 0. Fig 17 depicts the correlation distribution for the plain and encrypted images. The results indicate that the proposed cryptosystem can withstand correlation analysis.

The correlation coefficient is calculated using the formula (25) shown below.

$$r_{xy} = \frac{\sum_{i=1}^{N}\left(x_i - \frac{1}{N}\sum_{i=1}^{N}x_i\right)\sum_{i=1}^{N}\left(y_i - \frac{1}{N}\sum_{i=1}^{N}y_i\right)}{\sqrt{\sum_{i=1}^{N}\left(x_i - \frac{1}{N}\sum_{i=1}^{N}x_i\right)^2 \sum_{i=1}^{N}\left(y_i - \frac{1}{N}\sum_{i=1}^{N}y_i\right)^2}} \tag{25}$$

The intensity values of the two adjacent image pixels are denoted by $x$ and $y$. Fig 17a–17o displays the correlations discovered for both the unencrypted and original images. In the original image and the encrypted version, the correlations between the pixel boundaries are very strong and very low, respectively. The horizontal, vertical, and diagonal correlation coefficients for both the plain and encrypted images are summarized in Table 7. The proposed approach demonstrates strong encryption in terms of the degree of correlation between adjacent pixels, proving its good confusion and diffusion qualities.

## Histogram test

Histogram analysis is a security analysis method that reveals the statistical properties of ciphered images. In the histogram test, the encrypted image was divided into small blocks, and

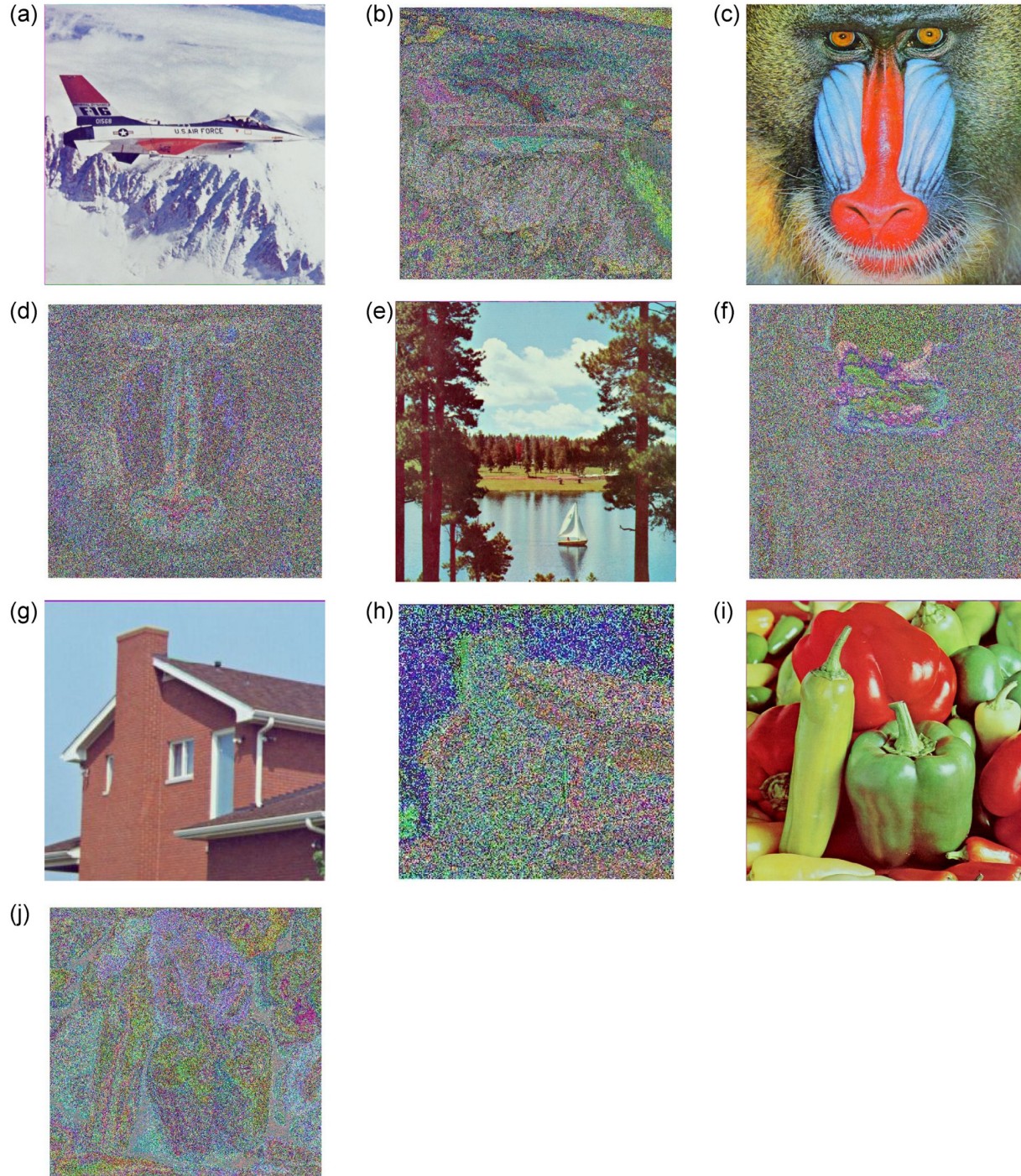

**Fig 16. Encryption decryption algorithm experimental data set images.** (**a**) Aeroplane (**b**) Aeroplane—Encryption (**c**) Baboon (**d**) Baboon—Encryption (**e**) Boat (**f**) Boat—Encryption (**g**) House (**h**) House—Encryption (**i**) Pepper (**j**) Pepper—Encryption.

**Table 7. Plain and cipher image correlation coefficients.**

| Image | Direction | | |
|---|---|---|---|
| | Horizontal | Vertical | Diagonal |
| Airplane | 0.1105 | 0.1105 | 0.6325 |
| Baboon | 0.0302 | 0.0302 | 0.5587 |
| Boat | 0.2174 | 0.2174 | 0.3526 |
| House | 0.0331 | 0.0331 | 0.5973 |
| Pepper | 0.0880 | 0.0880 | 0.4141 |

the frequency of each intensity level was counted and plotted as a histogram. The histogram of the encrypted image is then compared with the expected histogram of a random image. If the histogram of the encrypted image is close to the expected histogram, this suggests that the encryption process has added sufficient randomness to the image, and the encryption algorithm works as intended. Histogram tests can be useful in detecting certain types of encryption attacks, such as substitution attacks, in which the attacker substitutes one pixel value for another in the encrypted image. This can result in the histogram of the encrypted image deviating significantly from the expected histogram of a random image.

The histogram of a ciphered image shows how pixels in an image are distributed and is plotted by the number of pixels against the levels of color intensity. For a perfectly ciphered image, the histogram of the image must have a uniform distribution of pixels against the value of the color intensity. The proposed scheme and the designed method result in a uniform distribution. Therefore, the proposed scheme leaves no room for statistical attack.

A robust image cryptosystem approach should ensure the consistency of histograms for different cipher images. In Fig 18a–18j, the histograms of the plain images are contrasted with those of their equivalent cipher images, which are consistent with one another. Histograms of equivalents cipher images are similar to each other. The similarity of the histograms for the ciphered images is checked using a quantitative test known as variance (Var), expressed as provided in Eq (26).

$$Var(T) = \frac{1}{255^2} \sum_{p=0}^{255} \sum_{q=0}^{255} \frac{(t_p - t_q)^2}{2} \tag{26}$$

where $t_p$ and $t_q$ are pixel numbers whose gray values are $p$ and $q$, respectively, and $T = t_0, t_1, \cdots, t_{255}$ is the sequence of the histogram values. Table 8 displays the histogram variance results for the tested images before and after the encryption operation. Low numbers indicate a homogenous distribution in the histograms.

## Information entropy

Information entropy measures the degree of randomness of a particular message. In practice, the entropy of an encrypted image is determined by dividing it into segments and calculating the entropy of each segment. The probability of each intensity level in a block is calculated, and the entropy was computed using these data. This process was repeated for each block. The encryption algorithm may operate as intended, and the encrypted image may be secure if the entropy of the encrypted image is close to the maximum entropy for a certain image format. The entropy of a perfect grayscale image is eight.

The effectiveness of the proposed image cryptosystem was evaluated using an information entropy test on plain and analog cipher images. Table 9 displays the entropy results; the cipher

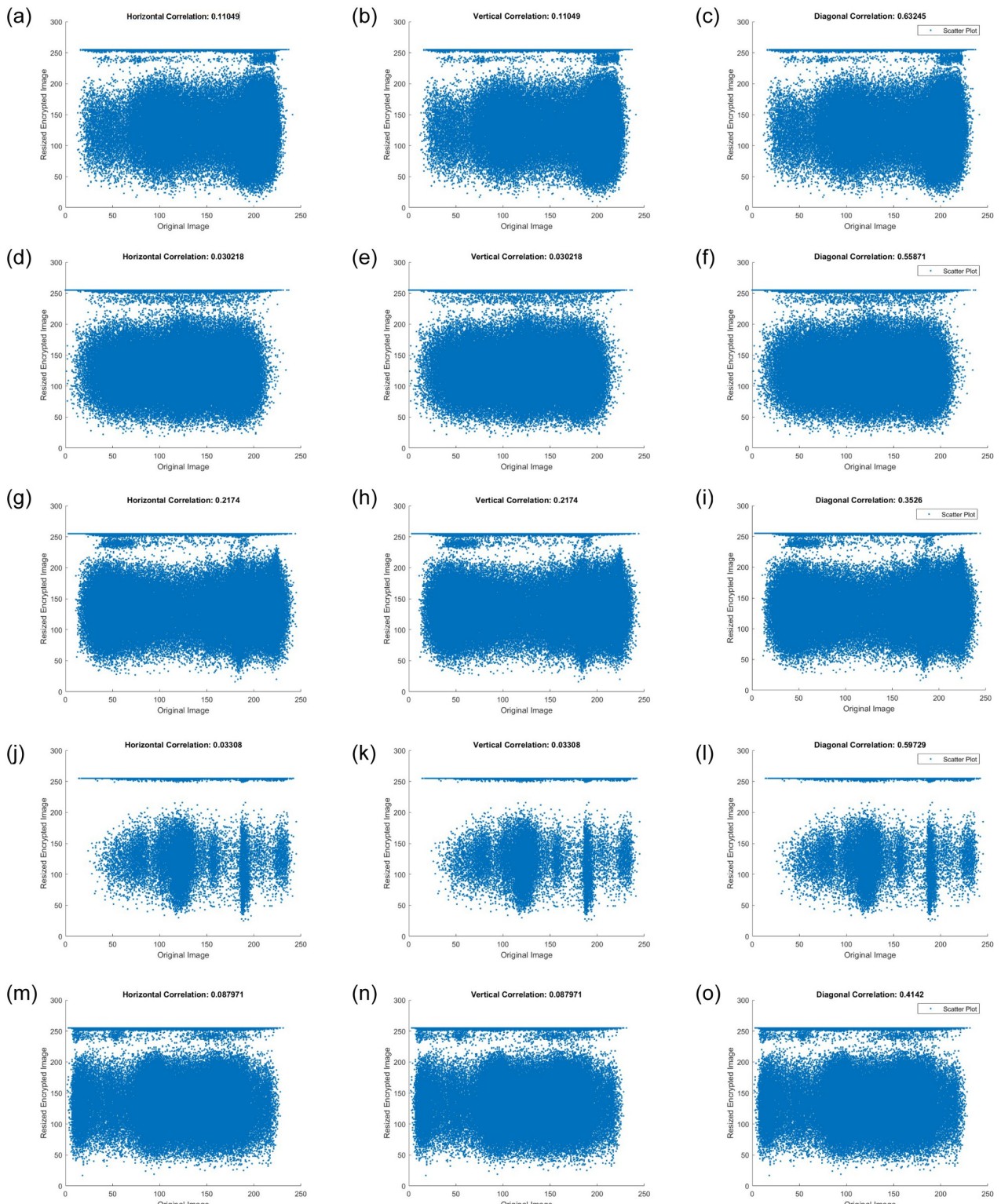

**Fig 17. Correlation distribution for plain and cipher images.** (**a**) Airplane (**b**) Pixel gray value on $(x, y)$—Plain Image (**c**) Pixel gray value on $(x, y)$—Cipher Image (**d**) Baboon (**e**) Pixel gray value on $(x, y)$—Plain Image (**f**) Pixel gray value on $(x, y)$—Cipher Image (**g**) Boat (**h**) Pixel gray value on $(x, y)$—Plain Image (**i**) Pixel gray value on $(x, y)$—Cipher Image (**j**) House (**k**) Pixel gray value on $(x, y)$—Plain Image (**l**) Pixel gray value on $(x, y)$—Cipher Image (**m**) Pepper (**n**) Pixel gray value on $(x, y)$—Plain Image (**o**) Pixel gray value on $(x, y)$—Cipher Image.

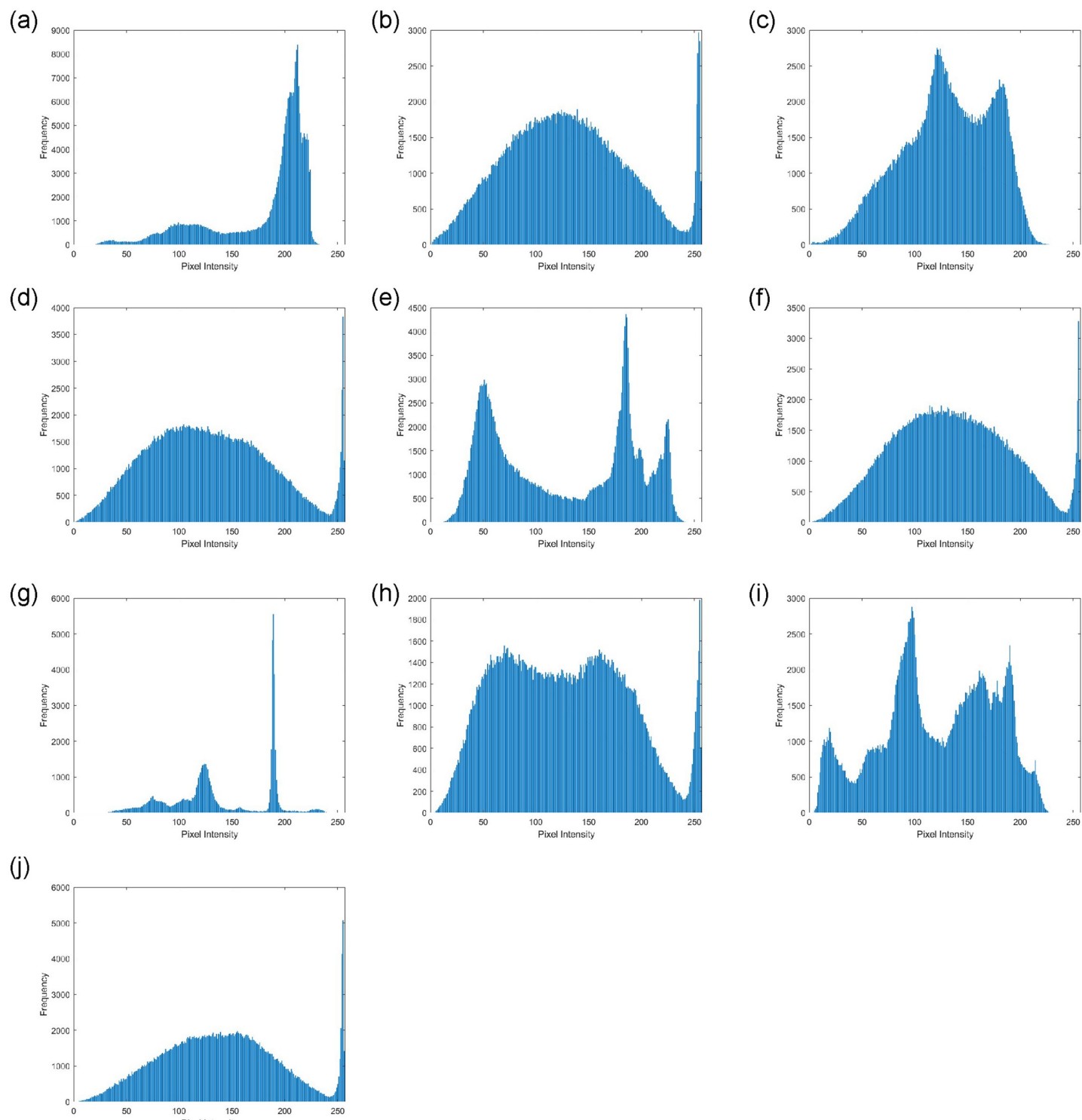

**Fig 18. Histogram analysis for plain and encrypted images.** (**a**) Histogram Airplane—Plain Image (**b**) Histogram Airplane—Cipher Image (**c**) Histogram Baboon—Plain Image (**d**) Histogram Baboon—Cipher Image (**e**) Histogram Boat—Plain Image (**f**) Histogram Boat—Cipher Image (**g**) Histogram House—Plain Image (**h**) Histogram House—Cipher Image (**i**) Histogram Pepper—Plain Image (**j**) Histogram Pepper—Cipher Image.

**Table 8. Histogram variance for plain and cipher images.**

| | Variance Value | |
|---|---|---|
| **Image** | **Plain** | **Cipher** |
| Airplane | 1942.50 | 3418.01 |
| Baboon | 3105.32 | 3332.27 |
| Boat | 1584.94 | 3125.49 |
| House | 2703.33 | 3730.28 |
| Pepper | 1329.86 | 3266.14 |

**Table 9. Entropy value for plain and cipher images.**

| | Information Entropy | |
|---|---|---|
| **Image** | **Plain** | **Cipher** |
| Airplane | 6.72 | 7.74 |
| Baboon | 7.37 | 7.73 |
| Boat | 7.47 | 7.71 |
| House | 6.47 | 7.78 |
| Pepper | 7.58 | 7.66 |

image entropy values were close to 8. These results demonstrate that the suggested cryptosystem is entropy-resistant.

## Differential analyses

Differential analysis is a powerful tool for breaking encryption algorithms, and has been successfully used to attack various encryption algorithms, including DES and RSA. To defend against differential analysis, encryption algorithms must be designed to resist such attacks, and thorough testing is necessary to ensure their security against the known types of differential attacks.

Any secure image cryptosystem must exhibit plain image sensitivity, allowing even the smallest changes to the plain image to significantly alter the cipher image. The Number of Pixels Change Rate(NPCR) and UACI tests were conducted to evaluate the proposed cryptosystem's plain image sensitivity (Unified Average Changing Intensity). They adhere to the following definition (27) and (28).

$$NPCR = \frac{\sum_{i=1}^{N} Diff(i)}{N} \times 100\% \tag{27}$$

$$\text{Diff}(i) = \begin{cases} 0 & \text{when } S_1(i) = S_2(i) \\ 1 & \text{when } S_1(i) \neq S_2(i) \end{cases}$$

$$UACI = \frac{1}{N} \left( \sum_{i=1}^{N} \frac{|S_1(i) - S_2(i)|}{255} \right) \times 100\% \tag{28}$$

where N stands for the complete pixel of the image, and $S_1$, $S_2$ are two cipher images created for a single plain image with minor changes to one of its bits. Table 10 presents the findings,

**Table 10. NPCR and UACI values.**

| | NPCR and UACI Values | |
|---|---|---|
| **Image** | **UACI** | **NPCR** |
| Airplane | 65.23% | 13.91% |
| Baboon | 65.28% | 14.16% |
| Boat | 65.31% | 15.06% |
| House | 65.21% | 13.90% |
| Pepper | 65.36% | 16.98% |

which demonstrate that the proposed cryptosystem is susceptible to minute changes in the plain image.

## Key space and key sensitivity

In the context of encryption, the terms "key space analysis" and "key sensitivity analysis" relate to methods for assessing the security of an encryption algorithm based on the size of the encryption key and the algorithm's sensitivity to changes in the encryption key. The phrase "key space" refers to the numerous keys that can be used in brute force attacks and should be sufficiently wide to prevent such attempts. Chaotic oscillator (1) is solved using the primary key parameters ($x_0$, $y_0$, $a$, $b$, $c$, $\mu$ $and$ $\omega$) in the encryption and decryption steps of the proposed encryption technique. Assuming that digital devices can calculate precisely, the key space for the provided cryptosystem is $10^{80}$, which is sufficient for any cryptographic technique.

Any secure cryptosystem must have a key sensitivity. Even slight changes in the key produce varying outcomes. The cipher images are repeatedly decrypted using minute adjustments to the key settings to gauge the key sensitivity of the proposed image cryptosystem. To quantitatively assess the key sensitivity, an NPCR test was conducted on the decrypted images using the real key and additional encrypted images using markedly different initial keys, as shown in Fig 19a–19j. Table 11 presents the results.

## Data loss attack

Cipher data may be vulnerable to data loss attacks when transmitted across a communication channel. Therefore, any image cryptosystem must be impervious to tracking users. To test the resistance of the suggested image cryptosystem to data loss attacks, Gaussian noise blocks were added to the cipher image, and an attempt was made to decrypt the corrupted cipher image to recover the secret data. The results of the data-loss attacks are shown in Fig 18, where the plain image may be successfully recovered from the deficient encryption image.

The peak signal-to-noise ratio (PSNR), mathematically expressed as Eq (29), is used to quantify the visual quality of the recovered images from defective cipher images.

$$
\begin{aligned}
PSNR(P, D) &= 20log_{10}\left(\frac{255}{\sqrt{MSE(P,D)}}\right) \\[2mm]
MSE(P, D) &= \frac{1}{x \times y}\sum_{i=1}^{x}\sum_{j=1}^{y}[P(i,j) - D(i,j)]^2
\end{aligned}
\tag{29}
$$

where D denotes the image recovered from the flawed image and $x{\times}y$ represents the dimensions of the plain image P. Table 12 shows the results of the PSNR test for the retrieved images. As observed from the results presented in Fig 20 and Table 12, the restored image loses more

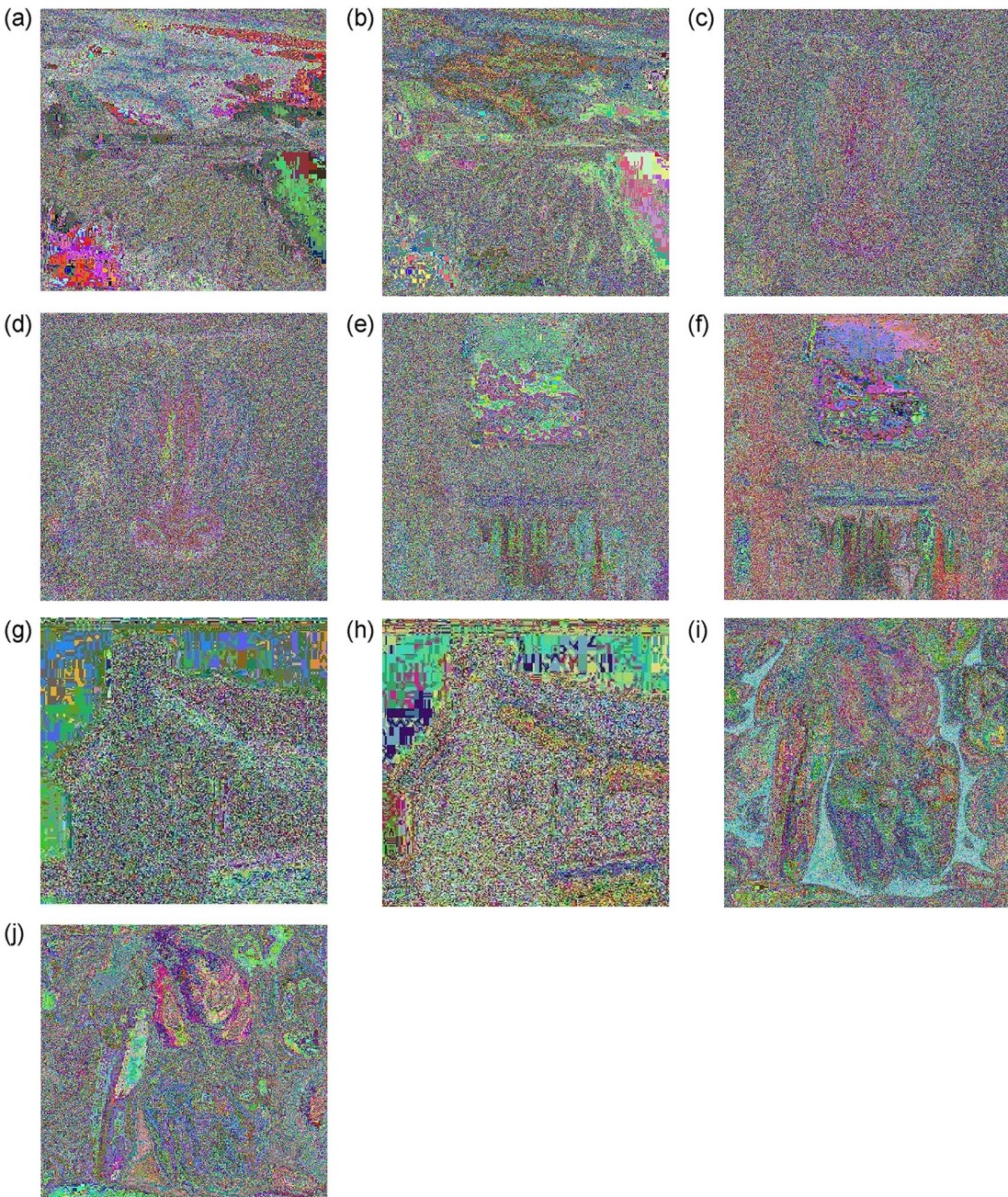

**Fig 19. Decryption effects of cipher images when making small changes in the key parameters.** (**a**) Airplane Key with ($x_0 = 0.1$, $y = 0.5$, $F = 2$) (**b**) Airplane Key with ($x_0 = 0.8$, $y = 0.5$, $F = 2.5$) (**c**) Baboon Key with ($x_0 = 0.8$, $y = 0.5$, $b = 1.3$) (**d**) Baboon Key with ($x_0 = 0.4$, $y = 0.1$, $F = 3.5$) (**e**) Boat Key with ($x_0 = 0.3$, $y = 0.8$, $b = 2.5$) (**f**) Boat Key with ($x_0 = 0.1$, $y = 0.5$, $F = 8$) (**g**) House Key with ($x_0 = 0.1$, $y = 0.5$, $F = 0.5$) (**h**) House Key with ($x_0 = 0.1$, $y = 0.1$, $b = 8.5$) (**i**) Pepper Key with ($x_0 = 1.5$, $y = 2.5$, $b = 10.5$) (**j**) Pepper Key with ($x_0 = 3.5$, $y = 2.5$, $F = 12$).

visual quality because more data are lost in the defective image. In Fig 18, which illustrates the consequences of decryption, the original image cannot be recovered, even after minute adjustments to the key parameters. The MSE and PSNR values for the original and decrypted images are listed in Table 13. This demonstrated the robustness of the proposed decryption algorithm.

**Table 11. Approximate encrypted pixels are different between the two encrypted Images.**

| Experimental Images | Key Sensitivity (%) |
|---|---|
| Aeroplane(Fig 19a and 19b) | 28.10% |
| Baboon (Fig 19c and 19d) | 28.23% |
| Boat(Fig 19e and 19f) | 29.07% |
| House (Fig 19e and 19f) | 28.32% |
| Pepper(Fig 19g and 19h) | 28.75% |

**Table 12. Lossing plain and cipher image MSE and PSNR details.**

| | Lossing Data Details | | | | | | | |
|---|---|---|---|---|---|---|---|---|
| Image | 20% | | 30% | | 40% | | 50% | |
| | MSE | PSNR (dB) | MSE | PSNR (dB) | MSE | PSNR (dB) | MSE | PSNR (dB) |
| Airplane | 387.14 | 22.25 | 575.35 | 20.53 | 764.81 | 19.30 | 952.62 | 18.34 |
| Baboon | 425.19 | 21.84 | 635.31 | 20.10 | 840.00 | 18.89 | 1049.35 | 17.92 |
| Boat | 413.77 | 21.96 | 620.62 | 20.20 | 822.62 | 18.98 | 1021.38 | 18.04 |
| House | 295.60 | 23.42 | 439.60 | 21.70 | 575.77 | 20.53 | 731.22 | 19.49 |
| Pepper | 384.71 | 22.28 | 569.40 | 20.58 | 753.96 | 19.36 | 941.38 | 18.39 |

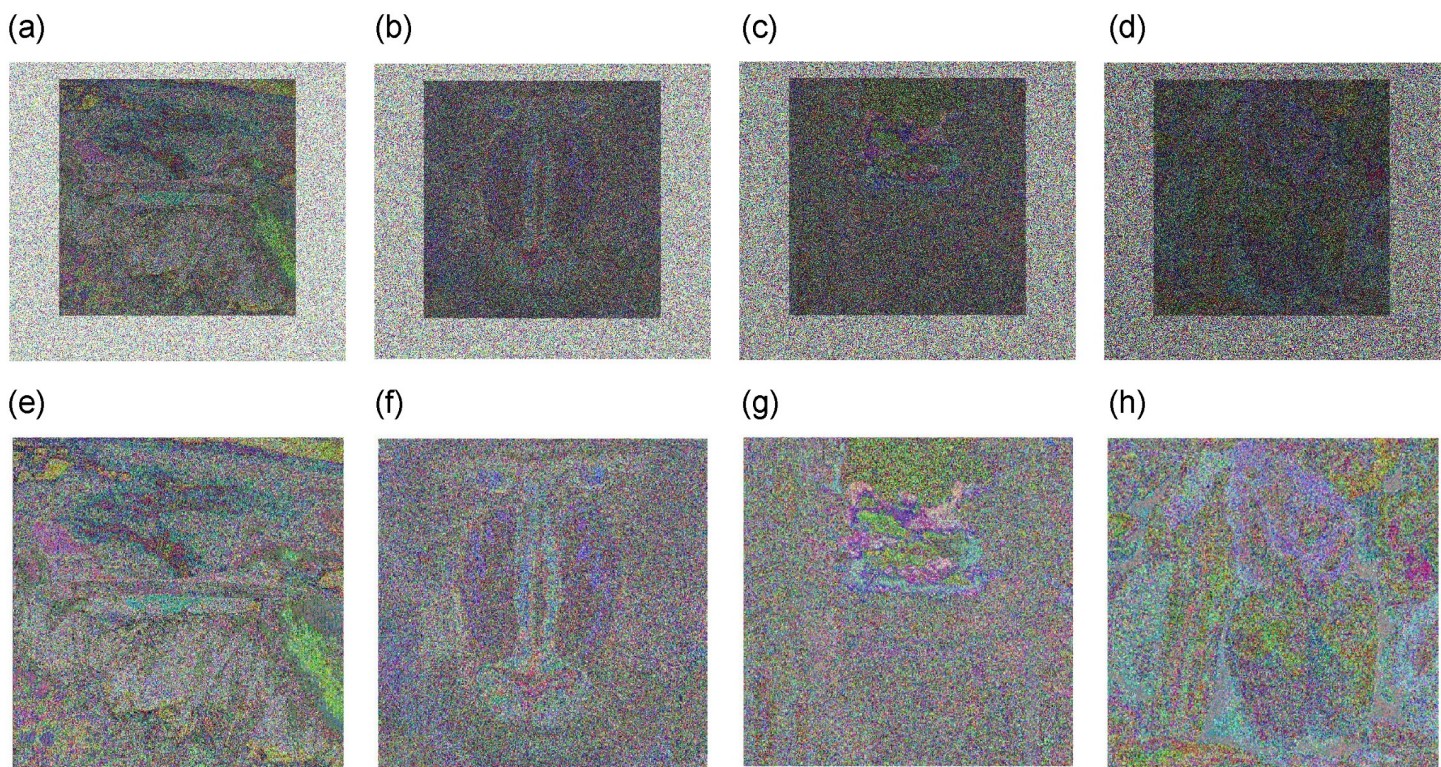

**Fig 20. Results of data loss attacks, retrieved effectively from the defective cipher image.** (**a**) Airplane Losing data by 20% (**b**) Baboon Losing data by 30% (**c**) Boat Losing data by 40% (**d**) Pepper Losing data by 50% (**e**) Airplane Retrieval data by 20% (**f**) Baboon Retrieval data by 30% (**g**) Boat Retrieval data by 40% (**h**) Pepper Retrieval data by 50%.

**Table 13. MSE and PSNR value for plain and decrypted images.**

| Image | MSE and PSNR | |
|---|---|---|
| | MSE | PSNR (dB) |
| Airplane | 2999.7 | 13.37 |
| Baboon | 2351.82 | 14.43 |
| Boat | 2480.9 | 14.31 |
| House | 2309.79 | 14.52 |
| Pepper | 2671.84 | 13.93 |

## Comparative reliability

Comparative reliability involves a meticulous evaluation of various cryptosystems to discern their consistency and operational performance, offering a nuanced understanding of the strengths and limitations inherent in each system.

The suggested encryption technique showcases several distinctive advantages:

1. Table 7 statistics lucidly demonstrate the proposed cryptosystem's resilience, maintaining cipher image correlation coefficients within the robust range of 0 to 1. This underscores its formidable resistance against correlation analysis.

2. Delving into Table 10 data unveils the exceptional sensitivity of the proposed cryptosystem, detecting even minute alterations within the original image.

3. The graphical representation in Fig 18 eloquently portrays the striking uniformity and coherence observed across the cipher image histograms, emphasizing their remarkable consistency.

4. A detailed examination of the results in Table 14 unequivocally establishes the steadfast adherence of the cipher images to histogram uniformity, further affirming the system's reliability.

5. A comprehensive analysis from Table 9 elegantly showcases that the cipher image entropy values are close to approximately 8, underscoring the system's robustness against entropy analysis.

6. Profound insights from Fig 19 and Table 11 emphatically highlight the high susceptibility of the proposed image encryption method to key variations, accentuating the significant impact that even minor key alterations have on the final output.

7. In-depth implications from Fig 20 and Table 12 vividly illustrate the direct correlation between data loss from the flawed image and the subsequent reduction in visual quality in the restored image.

**Table 14. Comparison of different image cryptosystems.**

| Image Cryptosystem | Comparison | | |
|---|---|---|---|
| | Information Entropy | UACI | Correlation |
| Proposed | 7.81 | 65.30% | 0.6604 |
| Xingyuan Wang et al. [52] | 7.25 | 64.24% | 0.0021 |
| Zhi-hua Gan et al. [53] | 7.79 | 65.24% | 0.0065 |
| Miguel Angel Murillo-Escobar et al. [26] | 7.69 | 65.29% | 0.0155 |
| Yulan Kang et al. [54] | 7.23 | 64.18% | 0.0028 |
| Ahmed A Abd El-Latif et al. [55] | 7.80 | 65.18% | 0.0013 |

8. The comprehensive findings in Table 13 incontrovertibly establish the robust correlation between unencrypted and encrypted images, affirming the encryption's unwavering reliability in preserving image integrity.

The highlighted core components facilitate a detailed comparative analysis, contrasting average measures such as information entropy, NPCR, UACI, and correlation coefficients of encryption systems with the established averages cited in [9, 28, 29, 45, 47]. These measures serve as essential yardsticks for assessing the effectiveness of the proposed image encryption method and provid a solid basis for comparison with other relevant techniques. The resulting comprehensive evaluation of the current system's performance is presented in Table 14, offering a discerning view within the framework of these well-accepted benchmarks.

The suggested text, including cryptographic algorithms and an image cryptographic algorithm, was incorporated into a variety of performance evaluations. This serves as significant evidence of the adaptability of the crypto system through the utilization of the designed chaotic circuit.

## Conclusion

This research involves constructing a circuit comprising a capacitor, resistor, inductor, and Chua's circuit. Utilizing existing models for the inductor, capacitor, and resistor, this study applies Kirchhoff's voltage-current law to derive the dynamic equations for the system. The dynamic properties of a novel chaotic oscillator were investigated, demonstrating its chaotic nature. Techniques such as bifurcation analysis, Lyapunov exponents, Poincaré maps, local Lyapunov exponent regions, periodicity, and resonance were employed to analyze the oscillator, while classical dynamics were used to examine the stability of the designed circuit. The analysis of Lyapunov exponents and bifurcation diagrams reveals that the chaotic system exhibits intricate dynamical behavior as the initial conditions and parameters change. Additionally, research on these systems has uncovered unique phenomena, including state transitions and the coexistence of attractors. Its diverse dynamic properties demonstrate its applicability to chaotic text and image cryptosystems.

The effectiveness of the proposed mechanisms was assessed using various methods, with results highlighting critical characteristics essential for reliable security purposes. Furthermore, a novel text and image cryptosystem is presented, utilizing the proposed chaotic sequence method along with the S-box approach. The efficiency of the recommended mechanisms has been evaluated through various assessments, indicating the essential qualities for ensuring trustworthy security. Moreover, the proposed encryption and decryption algorithms' outputs are compared with existing results, further reinforcing the effectiveness and robustness of the proposed chaotic oscillator circuit.

## Acknowledgments

The authors are grateful to the anonymous reviewers for their valuable comments.

## Author Contributions

**Conceptualization:** Aceng Sambas.

**Data curation:** Ibrahim Mohammed Sulaiman.

**Formal analysis:** Sathish Kumar Kumaravel, Ibrahim Mohammed Sulaiman.

**Investigation:** Suresh Rasappan, Basim A. Hassan.

**Methodology:** Sathish Kumar Kumaravel, Issam A. R. Moghrabi.

**Resources:** Basim A. Hassan.

**Software:** Suresh Rasappan, Sathish Kumar Kumaravel, Ibrahim Mohammed Sulaiman.

**Supervision:** Aceng Sambas.

**Validation:** Issam A. R. Moghrabi, Basim A. Hassan.

**Visualization:** Issam A. R. Moghrabi.

**Writing – original draft:** Suresh Rasappan.

**Writing – review & editing:** Aceng Sambas.

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
