## [Decision Letter · Decision Letter 0]

14 May 2024

PONE-D-24-08568A Novel Chua’s Based 2–D Chaotic System and Its Performance Analysis in CryptographyPLOS ONE

Dear Dr. Moghrabi,

Thank you for submitting your manuscript to PLOS ONE. After careful consideration, we feel that it has merit but does not fully meet PLOS ONE’s publication criteria as it currently stands. Therefore, we invite you to submit a revised version of the manuscript that addresses the points raised during the review process.

We look forward to receiving your revised manuscript.

Kind regards,

Xiaowei Li

Academic Editor

PLOS ONE

Journal Requirements:

5. Please ensure that you refer to Figure 1 in your text as, if accepted, production will need this reference to link the reader to the figure.

Reviewers' comments:

Reviewer's Responses to Questions

**Comments to the Author**

1. Is the manuscript technically sound, and do the data support the conclusions?

Reviewer #1: Yes

Reviewer #2: Yes

2. Has the statistical analysis been performed appropriately and rigorously? 

Reviewer #1: Yes

Reviewer #2: Yes

3. Have the authors made all data underlying the findings in their manuscript fully available?

Reviewer #1: Yes

Reviewer #2: Yes

4. Is the manuscript presented in an intelligible fashion and written in standard English?

Reviewer #1: Yes

Reviewer #2: Yes

5. Review Comments to the Author

Reviewer #1: The authors have expended their time, energy and knowledge to carry out this very robust research, which is not only interesting, but also contributes immensely to existing knowledge.

The work will however be more beautiful and easier to reference certain Mathematical equations, if all equations in the works are numbered rather than just numbering the 'important' model equations.

Reviewer #2: Comments: In this study, chaotic behavior has investigated in a second-order circuit with a nonlinear resistor and Chua's diode. Owing to the presence of a nonlinear capacitor and resistor among its components, this circuit has regarded as one of the simplest nonautonomous circuits. This study investigated numerous oscillator characteristics, highlighting their chaotic properties through bifurcations and Lyapunov exponents. The system exhibits a stable equilibrium point and chaotic attractor. The presented study is good. I recommend after revision.

1. Abstract is not clearly written as of scientific article.

2. Literature is poor and recent work need to be incorporated.

3. Include some applicability of Chua system to improve novelty strating from general dynamical system like: Haar wavelet approach to study the control of biological pest model in Tea plants, J Frac Calc & Nonlinear Sys (2023)4(2) : 14-30, Nabla Generalized Fractional Riemann-Liouville Calculus On Time Scales With An Application To Dynamic Equations, https://doi.org/10.48185/jfcns.v3i1.391, Study of Fractional Order Dynamical System of Viral Infection Disease under Piecewise Derivative." CMES-Computer Modeling in Engineering & Sciences 136.1 (2023). The Volterra-Lyapunov matrix theory and nonstandard finite difference scheme to study a dynamical system." Results in Physics 52 (2023): 106890.

4. There are many mistake related to grammer correct.

5. Slightly discuss existence of such system mathematically using the idea: On complex fractal-fractional order mathematical modeling of CO 2 emanations from energy sector." Physica Scripta 99.1 (2023): 015226.

6. Revise conclusion.

7. Check references some items are missing.

6. PLOS authors have the option to publish the peer review history of their article (what does this mean?). If published, this will include your full peer review and any attached files.

Reviewer #1: **Yes: **Moses Olayemi Adeyemi

Reviewer #2: No

---

## [Author Response · Author response to Decision Letter 0]

10 Jun 2024

Reviewer#1, Concern # 1 (The authors have expended their time, energy and knowledge to carry out this very robust research, which is not only interesting, but also contributes immensely to existing knowledge. The work will however be more beautiful and easier to reference certain Mathematical equations, if all equations in the works are numbered rather than just numbering the 'important' model equations.):

Author response: Thank you for your positive feedback and constructive suggestion. We appreciate your recognition of our efforts and the value of our research. In response to your recommendation, we will revise the manuscript to include numbering for all mathematical equations. This will enhance the clarity and ease of reference throughout the document. We believe that this adjustment will improve the overall readability and usefulness of our work for future researchers. Thank you for helping us to refine our paper.

Author action: We have revised the manuscript to include numbering for all mathematical equations, as per the reviewer's suggestion. This change has been implemented to improve the clarity and ease of referencing throughout the paper.__

Reviewer#2, Concern # 1 (Abstract is not clearly written as of scientific article.):

Author response: Thank you for your valuable feedback regarding the clarity of the abstract. We understand the importance of a well-written abstract in conveying the scope and significance of our research effectively.

Author action: The abstract has been revised to improve clarity and conciseness, ensuring it accurately reflects the scope, methodology, and key findings of the study in a manner consistent with scientific article standards.

Reviewer#2, Concern # 2 and Concern # 3 (Include some applicability of Chua system to improve novelty strating from general dynamical system like:

1. Haar wavelet approach to study the control of biological pest model in Tea plants, J Frac Calc & Nonlinear Sys (2023)4(2) : 14-30,

2. Nabla Generalized Fractional Riemann-Liouville Calculus On Time Scales With An Application To Dynamic Equations, https://doi.org/10.48185/jfcns.v3i1.391,

3. Study of Fractional Order Dynamical System of Viral Infection Disease under Piecewise Derivative." CMES-Computer Modeling in Engineering & Sciences 136.1 (2023).

4. The Volterra-Lyapunov matrix theory and nonstandard finite difference scheme to study a dynamical system." Results in Physics 52 (2023): 106890.):

Author response: Thank you for your insightful suggestion regarding the inclusion of the applicability of the Chua system to improve the novelty of our work. We appreciate the examples you provided and have carefully considered your feedback.

Our research, which includes a total of 52 references, already incorporates 23 articles published between 2020 and 2023. The structure of our study is divided into three parts: the design and qualitative analysis of the circuit, the implementation of backstepping control strategies, and the application of our findings in cryptography.

In response to your suggestion, we have included the following references to enhance the applicability and relevance of our work:

1. "Study of Fractional Order Dynamical System of Viral Infection Disease under Piecewise Derivative," CMES-Computer Modeling in Engineering & Sciences 136.1 (2023).

2. "The Volterra-Lyapunov matrix theory and nonstandard finite difference scheme to study a dynamical system," Results in Physics 52 (2023): 106890.

These additions help to contextualize the Chua system within broader applications and underline the novel aspects of our research.

Author action: We have revised the manuscript to include the suggested references, demonstrating the broader applicability of the Chua system. This enhancement aligns our work with contemporary research trends and highlights the innovative aspects of our study.________________________________________

Reviewer#2, Concern # 4 (There are many mistake related to grammer correct.):

Author response: Thank you for your feedback regarding grammatical errors in the manuscript. We understand the importance of clear and accurate language in effectively communicating our research.

Author action: We have utilized an AI tool, specifically Paperpal Copilot, to thoroughly review and correct grammatical errors throughout the manuscript. This ensures that the language is clear and adheres to high standards of academic writing.________________________________________

Reviewer #2, Concern # 5 and Concern # 7 (Slightly discuss existence of such system mathematically using the idea: 

On complex fractal-fractional order mathematical modeling of CO2 emanations from energy sector." Physica Scripta 99.1 (2023): 015226. 

Check references some items are missing):

Author response: Thank you for your insightful feedback and for suggesting the reference on complex fractal-fractional order mathematical modeling. While our research is primarily focused on differential equations, your suggestion has opened new avenues for considering fractional order modeling for this system.

Author action: We acknowledge the importance of fractional order modeling and are currently working on the numerical simulation part of this aspect, which we plan to include in our future work. Meanwhile, we have added the suggested reference,

1. "On complex fractal-fractional order mathematical modeling of CO2 emanations from energy sector," Physica Scripta 99.1 (2023): 015226, to our reference list.

 Additionally, we have reviewed and corrected any missing items in the references to ensure completeness and accuracy.

Reviewer#2, Concern # 6 (Revise conclusion):

Author response: Thank you for your suggestion to revise the conclusion of our manuscript. We appreciate your feedback and have carefully considered your input. 

Author action: Following your suggestion, we have modified the conclusion section of the manuscript to better encapsulate the key findings and implications of our research. The revised conclusion provides a more comprehensive summary of our study's contributions and highlights the significance of our findings in the broader context of the field. We believe that these changes enhance the overall clarity and impact of our paper.

Revised Conclusion:

This research involves constructing a circuit comprising a capacitor, resistor, inductor, and Chua's circuit. Utilizing existing models for the inductor, capacitor, and resistor, this study applies Kirchhoff’s voltage-current law to derive the dynamic equations for the system. The dynamic properties of a novel chaotic oscillator were investigated, demonstrating its chaotic nature. Techniques such as bifurcation analysis, Lyapunov exponents, Poincaré maps, local Lyapunov exponent regions, periodicity, and resonance were employed to analyze the oscillator, while classical dynamics were used to examine the stability of the designed circuit.

The analysis of Lyapunov exponents and bifurcation diagrams reveals that the chaotic system exhibits intricate dynamical behavior as the initial conditions and parameters change. Additionally, research on these systems has uncovered unique phenomena, including state transitions and the coexistence of attractors. Its diverse dynamic properties demonstrate its applicability to chaotic text and image cryptosystems.

The effectiveness of the proposed mechanisms was assessed using various methods, with results highlighting critical characteristics essential for reliable security purposes. Furthermore, a novel text and image cryptosystem is presented, utilizing the proposed chaotic sequence method along with the S-box approach. The efficiency of the recommended mechanisms has been evaluated through various assessments, indicating the essential qualities for ensuring trustworthy security. Moreover, the proposed encryption and decryption algorithms' outputs are compared with existing results, further reinforcing the effectiveness and robustness of the proposed chaotic oscillator circuit.

---

## [Decision Letter · Decision Letter 1]

24 Jun 2024

A Novel Chua’s Based 2–D Chaotic System and Its Performance Analysis in Cryptography

PONE-D-24-08568R1

Dear Dr. Moghrabi,

We’re pleased to inform you that your manuscript has been judged scientifically suitable for publication and will be formally accepted for publication once it meets all outstanding technical requirements.

Kind regards,

Xiaowei Li

Academic Editor

PLOS ONE

Additional Editor Comments (optional):

Reviewers' comments:

Reviewer's Responses to Questions

**Comments to the Author**

1. If the authors have adequately addressed your comments raised in a previous round of review and you feel that this manuscript is now acceptable for publication, you may indicate that here to bypass the “Comments to the Author” section, enter your conflict of interest statement in the “Confidential to Editor” section, and submit your "Accept" recommendation.

Reviewer #1: All comments have been addressed

Reviewer #2: (No Response)

2. Is the manuscript technically sound, and do the data support the conclusions?

Reviewer #1: Yes

Reviewer #2: Yes

3. Has the statistical analysis been performed appropriately and rigorously? 

Reviewer #1: Yes

Reviewer #2: (No Response)

4. Have the authors made all data underlying the findings in their manuscript fully available?

Reviewer #1: Yes

Reviewer #2: (No Response)

5. Is the manuscript presented in an intelligible fashion and written in standard English?

Reviewer #1: Yes

Reviewer #2: (No Response)

6. Review Comments to the Author

Reviewer #1: (No Response)

Reviewer #2: The revised version is acceptable. The introduction has been updated and recent related references have been added. The other comments also have been taken into account. I preferred if one at least of the following references have been added as was suggested in comment 3:

Haar wavelet approach to study the control of biological pest model in Tea plants, J Frac Calc & Nonlinear Sys (2023)4(2) : 14-30, Nabla Generalized Fractional Riemann-Liouville Calculus On Time Scales With An Application To Dynamic Equations, https://doi.org/10.48185/jfcns.v3i1.391

7. PLOS authors have the option to publish the peer review history of their article (what does this mean?). If published, this will include your full peer review and any attached files.

Reviewer #1: **Yes: **Moses Olayemi Adeyemi

Reviewer #2: No

---

## [Editor Report · Acceptance letter]

14 Aug 2024

PONE-D-24-08568R1 

PLOS ONE

Dear Dr. Moghrabi, 

I'm pleased to inform you that your manuscript has been deemed suitable for publication in PLOS ONE. Congratulations! Your manuscript is now being handed over to our production team.

Kind regards, 

on behalf of

Dr. Xiaowei Li 

Academic Editor

PLOS ONE